# Full-length MSP1 is a major target of protective immunity after controlled human malaria infection

Micha Rosenkranz[1] , Irene N Nkumama[2,3] , Rodney Ogwang[3], Sara Kraker[1] , Marie Blickling[1] , Kennedy Mwai[3,4],
Dennis Odera[3], James Tuju[3,5], Kristin Fürle[1] , Roland Frank[1], Emily Chepsat[3], Melissa C Kapulu[3],
CHMI-SIKA Study Team[3] , Faith HA Osier[3,6]

The merozoite surface protein 1 (MSP1) is the most abundant protein on the surface of the invasive merozoite stages of *Plasmodium falciparum* and has long been considered a key target of protective immunity. We used samples from a single controlled human malaria challenge study to test whether the full-length version of MSP1 (MSP1$_{FL}$) induced antibodies that mediated Fc-IgG functional activity in five independent assays. We found that anti-MSP1$_{FL}$ antibodies induced complement fixation via C1q, monocyte-mediated phagocytosis, neutrophil respiratory burst, and natural killer cell degranulation as well as IFNγ production. Activity in each of these assays was strongly associated with protection. The breadth of MSP1-specific Fc-mediated effector functions was more strongly associated with protection than the individual measures and closely mirrored what we have previously reported using the same assays against merozoites. Our findings suggest that MSP1$_{FL}$ is an important target of functional antibodies that contribute to a protective immune response against malaria.

## Introduction

Malaria remains a serious public health concern with ~249 million cases and ~608,000 deaths in 2022 (WHO, 2023). The malaria control toolbox has recently been boosted with two new vaccines, both targeting the circumsporozoite surface protein of the pre-erythrocytic sporozoite stage of the parasite. The first vaccine that was approved (RTS,S) confers 30–40% efficacy, with protection waning over the first year (RTSS Clinical Trials Partnership, 2015). The second vaccine (R21) was approved within 2 yr of the first and confers up to 75% efficacy with only modest waning over the first 12 mo (Datoo et al, 2024). However, as was observed in RTS,S phase III

studies, the efficacy of R21 varies by study site, with efficacy highest in areas with intense, seasonal as opposed to year round and high malaria transmission intensity. Furthermore, the requirement for repeated dosing over several years is challenging for implementation.

Epidemiological observations demonstrate that humans living in malaria-endemic regions naturally acquire immunity after repeated *Plasmodium falciparum* infections (Marsh & Kinyanjui, 2006). It is widely accepted that antibodies play a key role in antimalarial immunity (Cohen et al, 1961; Sabchareon et al, 1991) and emerging data highlight the importance of Fc-mediated effector functions for a protective immune response. This includes the recruitment of complement factors (Boyle et al, 2015; Reiling et al, 2019) natural killer (NK) cells (Odera et al, 2023), monocytes (Hill et al, 2013; Osier et al, 2014; Musasia et al, 2022), and neutrophils (Joos et al, 2010). We recently conducted a controlled human malaria infection study in semi-immune Kenyan adults (CHMI-SIKA) (Kapulu et al, 2022) where we showed that antibody-dependent phagocytosis of ring-stage parasites (Musasia et al, 2022) and activity of NK cells against merozoites (Odera et al, 2023) were important functional correlates of protection. Furthermore, additional analyses incorporating the full-panel of Fc-dependent and non-Fc mechanisms revealed that the breadth of effector functions targeting merozoites was most strongly associated with protection.

Two independent lines of evidence led us to investigate full-length merozoite surface protein 1 (MSP1$_{FL}$) as a target of important Fc-mediated mechanisms. First, it is the most abundant protein on the merozoite surface (Gilson et al, 2006) and we have recently shown that the breadth of effector functions was strongly correlated with antibody binding to whole merozoites (Nkumama et al, 2022 *Preprint*). Second, an independent and recent vaccine trial using MSP1$_{FL}$ showed that the vaccine was safe, highly immunogenic and induced opsonizing antibodies that stimulated Fc-mediated activity in multiple assays (Blank et al, 2020; Rosenkranz et al, 2023). MSP1 is expressed as a ~190-kD precursor protein that is enzymatically processed by subtilisin-like

[1]Centre of Infectious Diseases, Heidelberg University Hospital, Heidelberg, Germany   [2]B Cell Immunology, German Cancer Research Centre, Heidelberg, Germany   [3]Centre for Geographic Medicine Research (Coast), Kenya Medical Research Institute-Wellcome Trust Research Programme, Kilifi, Kenya   [4]Epidemiology and Biostatistics Division, School of Public Health, University of the Witwatersrand, Johannesburg, South Africa   [5]Department of Biotechnology and Biochemistry, Pwani University, Kilifi, Kenya   [6]Department of Life Sciences, Imperial College London, London, UK

Correspondence: f.osier@imperial.ac.uk; micha.rosenkranz@web.de
A full list of the Controlled Human Malaria Infection in Semi-Immune Kenyan Adults (CHMI-SIKA) study team and their affiliations appear at the end of the paper.

proteases (SUB-1) generating four major subunits, p83, p30, p38, and p42. The subunits remain non-covalently attached to each other and are tethered to the plasma membrane via a glycosylphosphatidylinositol anchor. During merozoite invasion, SUB-2 cleaves C-terminal p42 resulting in the formation of p33 and p19; the letter gets internalized during invasion, whereas the rest of the protein is shed (Blackman & Holder, 1992). Antibodies targeting various regions of MSP1 have been shown to promote several immune mechanisms, including the direct inhibition of parasite growth by blocking RBC invasion (Blackman et al, 1990; Woehlbier et al, 2006; Woehlbier et al, 2010) or inducing Fc-mediated effector functions such as complement fixation (Boyle et al, 2015), opsonic phagocytosis activity (OPA) of monocytes (Kana et al, 2019), and respiratory burst of neutrophils (Joos et al, 2015; Jaschke et al, 2017).

Whereas MSP1 based vaccines showed protective efficacy in animal models (Perrin et al, 1984; Herrera et al, 1990; Etlinger et al, 1991), clinical trials in humans have been disappointing overall. Importantly, all human trials focused on subunits of MSP1 such as the p42 subunit rather than the full-length molecule and therefore missed ~80% of the protein and important epitopes that might be relevant for protection (Ogutu et al, 2009; Sheehy et al, 2012). The evidence that antibodies targeting MSP1 are important for naturally acquired immunity has been conflicting. This is in part because of differences in the subunit, allelic form, and quality of the protein that was assessed (al-Yaman et al, 1996; Egan et al, 1996; Dodoo et al, 1999; Nebie et al, 2008; Osier et al, 2008; Richards et al, 2013). Most sero-epidemiological studies focused on subunits of the full-length MSP1 molecule from either the conserved C-terminal (p19 or p42) (Okech et al, 2004; Wilson et al, 2011) or the polymorphic N-terminal domain (MSP1 Block 2) (Cavanagh et al, 1998; Cavanagh et al, 2004).

These analyses of MSP1 were conducted in traditional cohort studies that have important limitations in the assessment of naturally acquired immunity. First, the definition of clinical episodes of malaria is often confounded by the presence of asymptomatic parasitaemia, leading to a misclassification bias (Kinyanjui et al, 2009). This is compounded by an inability to confirm that children who did not develop a clinical episode of malaria during a period of observation were actually challenged with an infection (Marsh & Kinyanjui, 2006). Second, the timing, strain, and infecting dose of parasites is not controlled and assumed to be even across the study population. Third, the onset of malaria illnesses is difficult to verify and relies on parental call. These limitations are largely mitigated in controlled human challenge infections, where the infection parameters are quantified, asymptomatic parasitaemia can be cleared at the start of the study and the risk of further unplanned infections can be minimized. Participant follow-up for clinical symptoms is documented accurately as soon as it occurs as they are accommodated in residential study facilities for the duration of the study (Kapulu et al, 2018).

We leveraged this human challenge platform to test whether antibodies to $MSP1_{FL}$-induced IgG Fc-mediated function in multiple assays, as had previously been observed with merozoites (Nkumama et al, 2022 Preprint). The challenge study was named CHMI-SIKA: Controlled Human Challenge Infection in Semi-Immune

Kenyan Adults. We found that volunteers who were protected from sporozoite challenge had high levels of anti-$MSP1_{FL}$ antibodies and induced $MSP1_{FL}$-specific Fc-mediated effector function in five distinct assays involving complement, neutrophils, natural killer cells and phagocytes.

# Results

## CHMI-SIKA study outcomes

Of the 142 study participants who were included in the final analysis, a proportion were treated before the end of the study on day 22 (39%, 56/142), whereas the remainder were not (61%, 86/142), referred to as "treated" and "non-treated," respectively. Treatment before the end of the study was provided if the volunteers developed clinical symptoms of malaria, including a fever (sub-classified as febrile, n = 26) or if there were no clinical symptoms but parasitaemia exceeded a predefined threshold of 500 parasites/$\mu$l (subclassified as non-febrile). Artemether-Lumefantrine was used for treatment. Volunteers not requiring treatment before the end of the study were subclassified into those that were PCR positive for blood stage malaria parasites (n = 53) and those that were PCR negative (n = 33). At the end of the study and before discharge on day 22, all volunteers that remained free of clinical symptoms were treated to ensure that all infections were cleared (Kapulu et al, 2022).

## Anti-MSP1$_{FL}$ antibody levels were higher in non-treated versus treated CHMI-SIKA volunteers

To test the potential relevance of $MSP1_{FL}$ in protection from malaria, we assessed $MSP1_{FL}$-specific IgG, IgM, and IgG subclass antibodies in plasma samples from CHMI-SIKA volunteers (n = 142) that were collected one day before sporozoite challenge (C-1). Interestingly, we found that the seroprevalence of IgG was comparable in treated and non-treated volunteers (95% versus 100%, respectively), whereas IgM was predominant in non-treated volunteers (69% versus 20%, respectively, Fig 1A).

We investigated IgG subclasses and found that the response was dominated by cytophilic IgG1 (82%:100%, treated versus non-treated) and IgG3 (86%:98%, treated versus non-treated), whereas non-cytophilic IgG2 (16%:38%, treated versus non-treated) and IgG4 (25%:62%, treated versus non-treated) antibodies were less abundant (Fig 1A). Within clinical subgroups, non-treated volunteers that were either PCR+ or PCR− tended to have a higher prevalence of anti-$MSP1_{FL}$ antibodies compared with those that were treated and either febrile or non-febrile. However, the overall difference for total IgG and cytophilic IgG between subgroups was small (Fig 1B).

In marked contrast to the antibody prevalence, anti-$MSP1_{FL}$ IgG, IgM and IgG subclass antibody levels were significantly higher in non-treated versus treated volunteers (P < 0.0001, Fig 1C). Within subgroups, both the non-treated PCR+ and PCR− volunteers had significantly higher anti-$MSP1_{FL}$ antibody levels than those that were treated and either febrile or non-febrile (P < 0.0001–0.0213, Fig 1D).

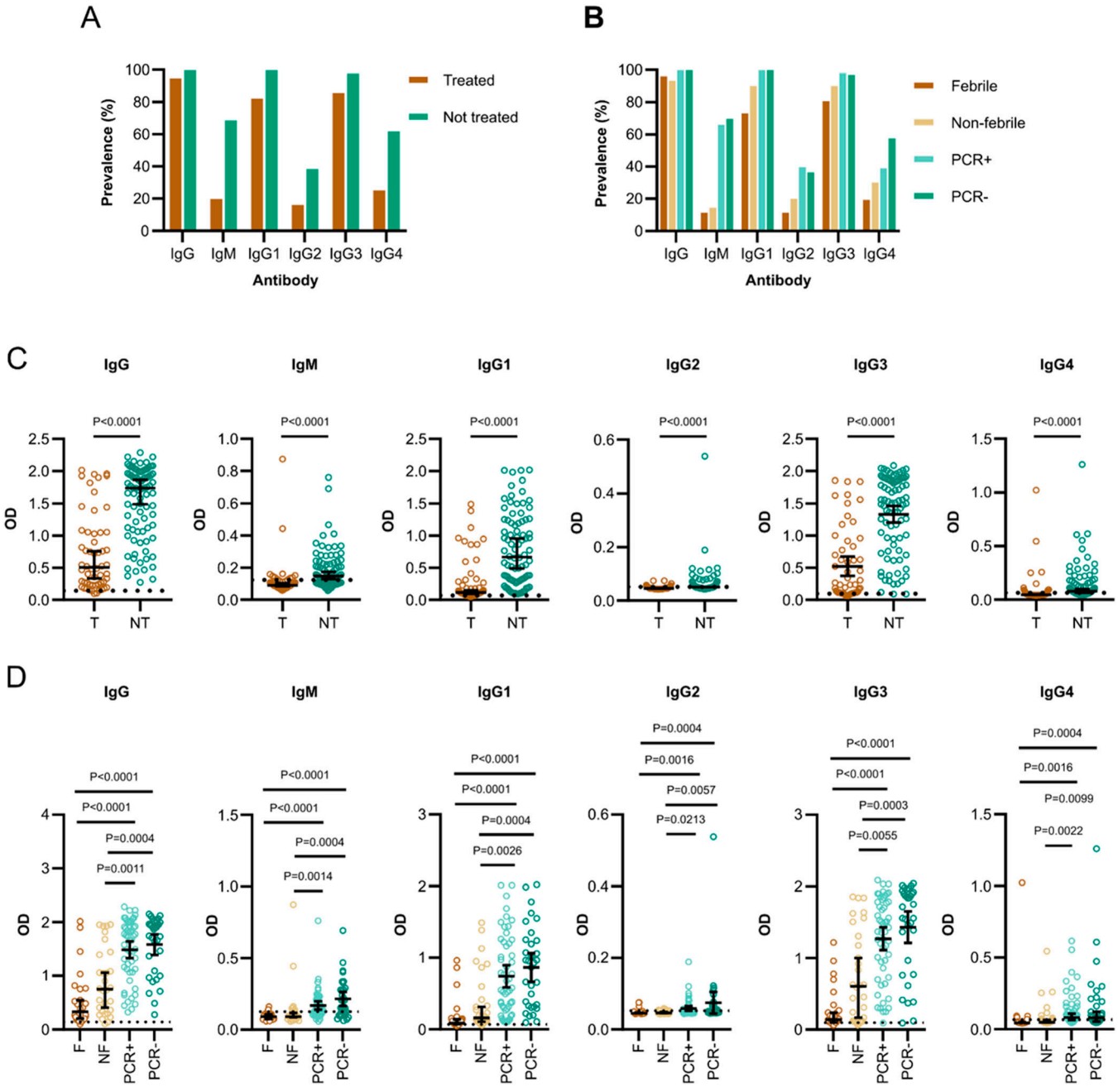

**Figure 1. High antibody levels measured by ELISA against MSP1$_{FL}$ in volunteers who were protected from sporozoite challenge.**
**(A)** The prevalence of IgG, IgM, and IgG subclass 1–4 antibodies in treated (T, n = 56) and non-treated volunteers (NT, n = 86). **(B)** The prevalence of IgG, IgM, and IgG subclass 1–4 antibodies compared between the subgroups based on parasite growth patterns, treated febrile (F, n = 26), treated non-febrile (NF, n = 30), non-treated PCR positive (PCR+, n = 53), and non-treated PCR negative (PCR−, n = 33). **(C)** Levels of IgG, IgM, and IgG subclass 1–4 antibodies compared between treated and non-treated volunteers. **(D)** Levels of IgG, IgM, and IgG subclass 1–4 antibodies were compared between the four different subgroups. Each data point represents antibody levels for one sample in duplicates measured in one experiment. The seropositivity cut-off value was calculated as the optical density (OD) of malaria-naïve plasma samples (n = 5) plus three standard deviations indicated as the dotted line. Error bars represent the median plus 95% confidence intervals. Statistical differences between treatment outcomes were calculated using the Mann-Whitney test and between subgroups using the Kruskal-Wallis test with Dunn's multiple comparisons test.

Next, we compared the magnitude of the responses between antibody isotypes and observed that IgG levels were significantly higher ($P < 0.0001$) than IgM (Fig S1A). Similarly, the levels of cytophilic IgG1 and IgG3 subclass antibodies were similarly higher ($P < 0.0001$) than the non-cytophilic IgG2 and IgG4 (Fig S1B); however,

caution should be taken when comparing the magnitude of responses between the antibody classes because of potential differences in the sensitivity of secondary antibodies.

We observed significant and positive correlations between antibody levels against MSP1$_{FL}$ and merozoites (Fig S2) suggesting co

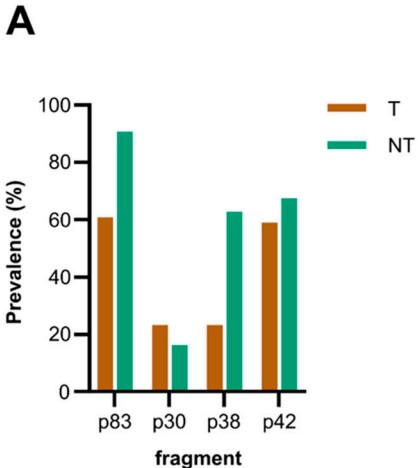

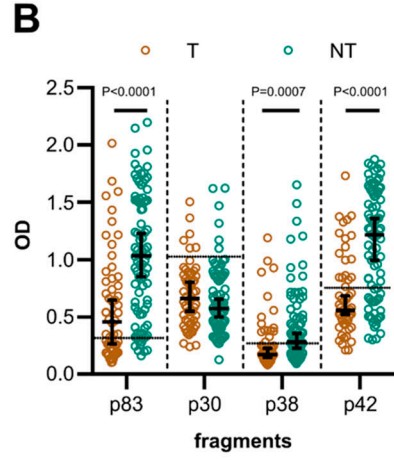

**Figure 2.  Antibody levels against subunits were high in non-treated volunteers and cross-reactive.**
**(A)** The prevalence of IgG antibodies in treated (T, n = 56) and non-treated volunteers (NT, n = 86). **(B)** IgG antibody levels against MSP1 subunits were compared between treated (T, n = 56) and non-treated volunteers (NT, n = 86). **(C)** Spearman's correlation of anti-MSP1$_{FL}$ (3D7) and MSP1$_{FL}$-F (K1) antibody levels (OD) for CHMI volunteers (n = 142). **(D)** IgG antibody levels against MSP1$_{FL}$-F were compared between treated (T, n = 56) and non-treated volunteers (NT, n = 86). Each data point represents antibody levels for one sample in duplicates measured in one experiment. The seropositivity cut-off value was calculated as the optical density (OD) of malaria-naïve plasma samples (n = 5) plus three standard deviations indicated as the dotted line. Error bars represent the median plus 95% confidence intervals. Statistical differences between treatment outcomes were calculated using the Mann-Whitney test.

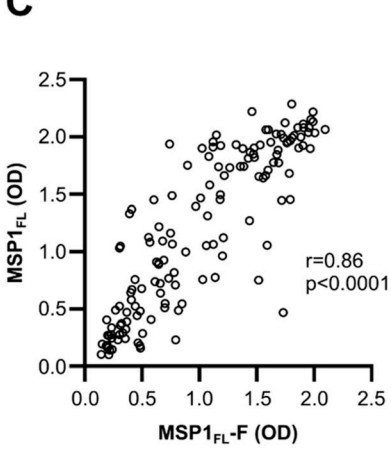

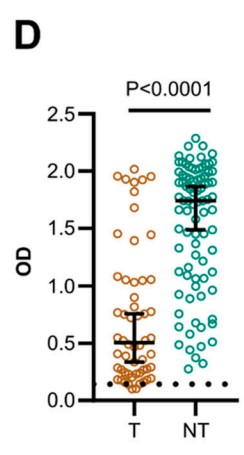

acquisition of anti-MSP1$_{FL}$ antibodies alongside antibodies against other merozoite antigens. The strongest correlations were observed for cytophilic IgG1 ($r$ = 0.81, CI 0.75–0.86, $P < 0.0001$) and IgG3 ($r$ = 0.82, CI 0.76–0.87, $P < 0.0001$), whereas for non-cytophilic IgG2 and IgG4 antibodies, only moderate correlations were detected ($r$ = 0.46–0.59, $P < 0.0001$).

### IgG antibodies detect each subunit of MSP1$_{FL}$ and are cross-reactive

We next investigated whether the IgG response was preferentially directed towards specific regions of the MSP1 molecule. We compared responses against the p83, p30, p38, and p42 subunits of MSP1 in non-treated and treated CHMI volunteers. As shown in Fig 2A, we observed antibody responses directed against all MSP1 subunits, with the highest prevalence against the p83 subunit, followed by p42, p38, and p30. The prevalence of antibodies was significantly higher in non-treated compared with treated volunteers for the p83 and p38 subunits (91% versus 61% and 63% versus 23%, respectively, $P < 0.0001$) but not for p30 and p42 (67% versus 59% and 16% versus 23%, $P$ = 0.332 and 0.296, respectively).

In line with the previously reported antibody responses against full-length MSP1, the antibody levels against p83, p38, and p42 were significantly higher ($P < 0.0001$–0.0007, Fig 2B) for non-treated volunteers compared with treated individuals. However, for p30, no significant difference between the groups was observed. Important to note, for p30 and p42 we also observed higher background signals (OD = 0.7–1.0) compared with p38 and p83 (OD = 0.3), potentially because of antibody cross-reactivities from malaria-naïve adults. Despite the different background intensities, the immunogenicity profiles are comparable with previous results from semi-immune adults living in Burkina Faso (Woehlbier et al, 2006).

MSP1 exists in two main allelic forms, represented by the MAD20 and K1 variant (Tanabe et al, 1987). Our ELISA assays used MSP1$_{FL}$ based on the *P. falciparum* 3D7 strain that is like the MAD20 variant at this locus. We tested whether the antibodies were cross-reactive with MSP1$_{FL}$-F that is based on the K1 variant. We found a high correlation between both variants ($r$ = 0.86, 95% CI 0.81–0.90, $P < 0.0001$ [Spearman's rho], Fig 2C). Not surprisingly, therefore, non-treated volunteers had significantly higher IgG levels against MSP1$_{FL}$-F compared with treated volunteers ($P < 0.0001$, Fig 2D) which implies that antibodies bind to conserved and/or dimorphic regions.

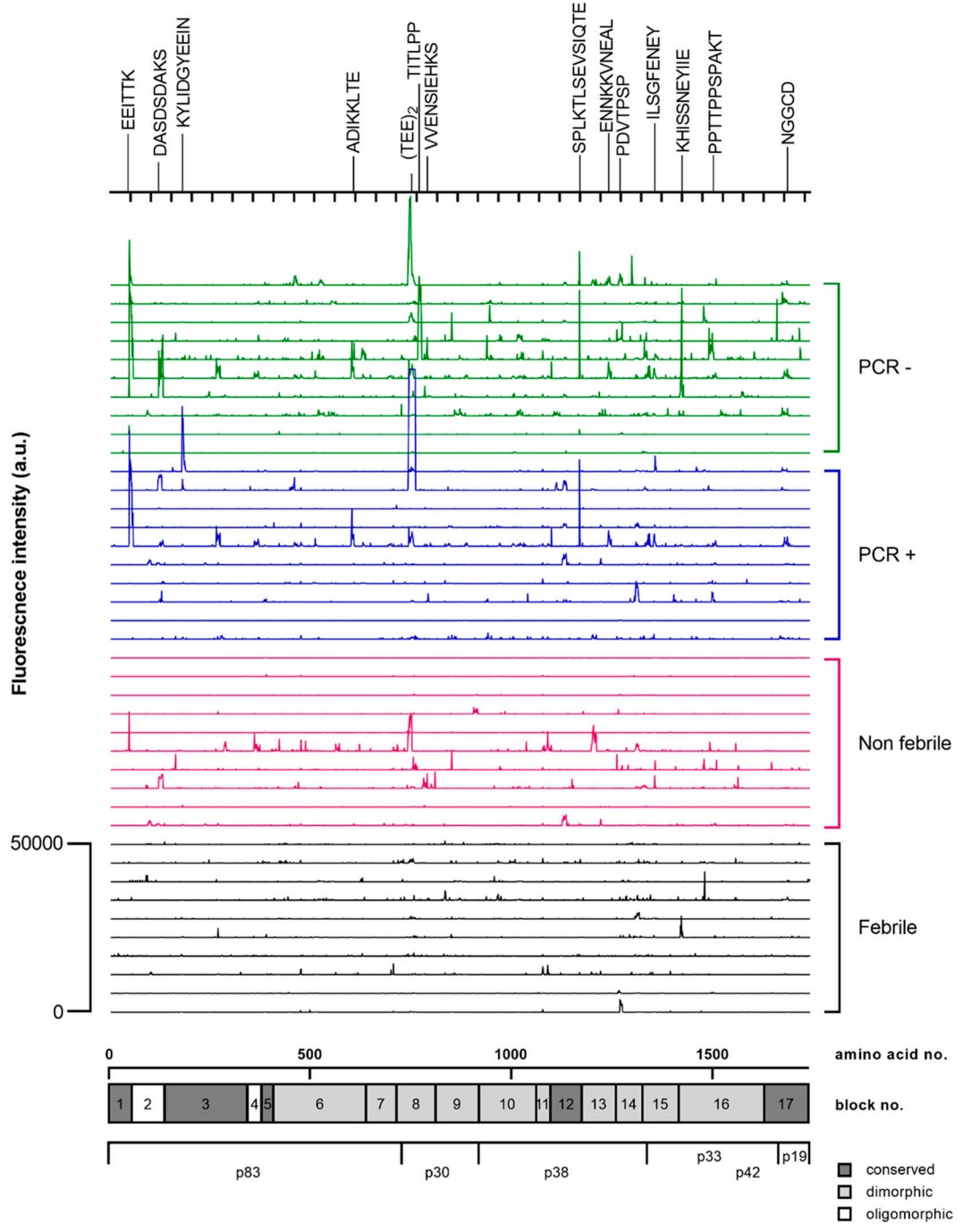

**Figure 3. IgG from treated volunteers bind to conserved and dimorphic epitopes across the whole MSP1 molecule.**
Fluorescence intensity landscapes across MSP1$_{FL}$ shown for individuals of the four subgroups based on parasite growth patterns: treated febrile (n = 10), treated non-febrile (n = 10), non-treated PCR positive (n = 10), and non-treated PCR negative (n = 10). Every line represents a sample. Relevant epitopes have been highlighted on top. A graphical representation of the primary structure of MSP1$_{FL}$ is shown below the fluorescence intensity landscapes.

## IgG from non-treated volunteers binds to conserved and dimorphic epitopes across the MSP1$_{FL}$ molecule

To better localize antibody binding, we mapped linear B-cell epitopes using an MSP1$_{FL}$ (3D7) peptide chip consisting of 1,715 15-mer peptides with a peptide–peptide overlap of 14 amino acids. We randomly selected 10 plasma samples from each clinical subgroup and analysed their linear epitope repertoire. As shown in Fig 3, we identified numerous epitopes across the entire MSP1$_{FL}$ molecule that induce an IgG response. Dominant epitopes were found in the conserved N- and C-terminal regions (EEITTK, position 56–61, p83 and SPLKTLSEVSIQTE, position 1,150–1,163, p38) and central

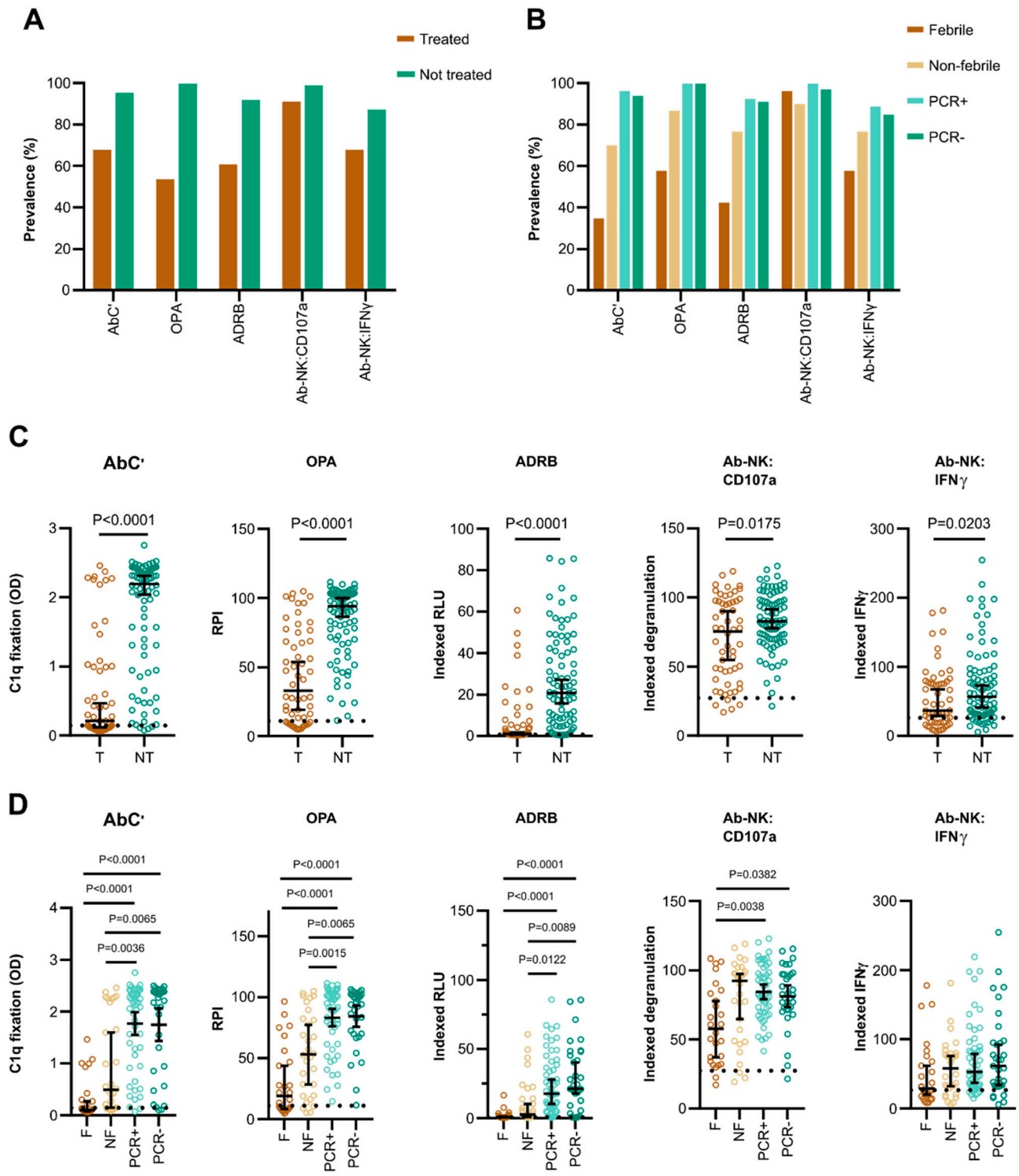

**Figure 4. Prevalence and magnitude of MSP1$_{FL}$-specific Fc-effector functions are high in non-treated versus treated volunteers.**
**(A)** The prevalence of MSP1$_{FL}$-specific Fc-mediated functional activities in treated (T, n = 56) and non-treated volunteers (NT, n = 86). **(B)** The prevalence of MSP1$_{FL}$-specific Fc-mediated functional activities compared between the subgroups based on parasite growth patterns, treated febrile (F, n = 26), treated non-febrile (NF, n = 30), untreated PCR positive (PCR+, n = 53), and untreated PCR negative (PCR−, n = 33). **(C)** Levels of Fc-mediated effector functions of anti-MSP1$_{FL}$ antibodies were compared between treated and non-treated volunteers. **(D)** Levels of Fc-mediated effector functions of anti-MSP1$_{FL}$ antibodies were compared between the different subgroups, based on parasite growth densities: treated febrile, treated non-febrile, non-treated PCR positive, and non-treated PCR negative. Each data point represents antibody levels for one sample in duplicates measured in one experiment. The seropositivity cut-off value was calculated as the activity level of malaria-naïve plasma samples (n = 5) plus three standard deviations indicated as the dotted line. Error bars represent the median plus 95% confidence intervals. Statistical differences between treatment outcomes were calculated using the Mann-Whitney test and between subgroups using the Kruskal-Wallis test with Dunn's multiple comparisons test. CD107a,

dimorphic domains (ETEETEET, position 747–754, p30). The most treated febrile and treated non-febrile volunteers showed limited binding events and non-treated PCR+ and PCR– individuals showed significantly stronger responses to several conserved and dimorphic regions compared with treated volunteers (Table S1).

### Anti-MSP1$_{FL}$ antibodies induce high levels of Fc-mediated effector functions that are associated with protection

We next investigated whether MSP1$_{FL}$ was a target of opsonizing antibodies that could induce Fc-mediated effector mechanisms. Therefore, we used four antigen-specific in vitro functional assays to measure antibody-dependent complement fixation (AbC') (Boyle et al, 2015; Reiling et al, 2019), OPA of MSP1$_{FL}$-coupled fluorescent beads by monocytes (Kana et al, 2019), antibody-dependant respiratory burst (ADRB) of neutrophils (Joos et al, 2015), and antibody-dependent natural killer cell (Ab-NK) activities (Odera et al, 2023)The latter was assessed by multiparameter flowcytometry resulting in two readouts: degranulation of NK cells (Ab-NK: CD107a) and IFNγ production (Ab-NK:IFNγ).

We compared MSP1$_{FL}$-specific Fc-mediated effector functions between volunteers who required treatment and those who did not. We found a higher prevalence of functional antibodies across all five effector mechanisms for non-treated compared with treated volunteers (Fig 4A and B). Similarly, the magnitude of Fc-mediated functions was significantly higher for non-treated volunteers compared with those who were treated (AbC'; $P < 0.0001$, OPA; $P < 0.0001$, ADRB; $P < 0.0001$; Ab-NK:CD107a; $P = 0.0175$, Ab-NK:IFNγ; $P = 0.0203$, Fig 4C). We also observed that non-treated PCR+ and PCR– individuals had significantly higher AbC', OPA, and ADRB activity compared with treated febrile or non-febrile individuals. For Ab-NK activities, we only observed significant differences in degranulation (Ab-NK:CD107a) for non-treated PCR ± versus treated febrile individuals but IFNγ production (Ab-NK:IFNγ) was not significantly different across the subgroups (Fig 4D).

We further explored the potential relationship between Fc-mediated effector functions and antibody levels and observed high correlations between anti-MSP1$_{FL}$ IgG and AbC' ($r = 0.83$, 95% CI 0.77–0.88, $P < 0.0001$), ADRB ($r = 0.91$, 95% CI 0.88–0.94, $P < 0.0001$), and OPA ($r = 0.89$, 95% CI 0.85–0.92, $P < 0.0001$, Fig 5A). For both readouts of the Ab-NK assay, we observed moderate ($r > 0.5$) correlations with IgG. Correlations with IgM were low with Ab-NK ($r > 0.2$), and moderate ($r > 0.5$) with AbC' ($r = 0.58$, 95% CI 0.46–0.69, $P < 0.0001$), ADRB ($r = 0.60$, 95% CI 0.48–0.70, $P < 0.0001$), and OPA ($r = 0.55$, 95% CI 0.42–0.66, $P < 0.0001$). Cytophilic antibodies (IgG1 and IgG3) can efficiently bind complement and most of the Fc-receptors (FcRs) of immune cells (Vidarsson et al, 2014). As expected, the correlations between cytophilic antibodies and effector functions were higher compared with non-cytophilic antibodies ($r = 0.43–0.90$ versus 0.29–0.66, respectively). Next, the correlations between effector functions were explored (Fig 5A). OPA, ADRB, and AbC' were highly correlated with each other ($r = 0.86–0.92$, $P < 0.0001$) but less so for Ab-NK ($r = 0.47–0.59$, $P < 0.0001$). As expected, the two readouts

of the Ab-NK assay (CD107a and IFNγ) were highly correlated ($r = 0.88$, 95% CI 0.84–0.91, $P < 0.0001$) with each other.

To identify MSP1$_{FL}$-specific effector functions that may be important for protection against malaria in the CHMI study, we analysed the association of each effector function individually with the time to treatment during follow-up. To do this, responses were converted into two categories (high and low) based on function-specific thresholds using maximally selected rank statistics (Musasia et al, 2022; Nkumama et al, 2022 Preprint; Odera et al, 2023). Cox proportional hazards were adjusted for residual lumefantrine levels and year of study as confounders. Interestingly, we observed statistically significant and strong associations with protection for each MSP1$_{FL}$-specific effector function with aHR estimates ranging between 0.15–0.35 (Fig 5B).

### The breadth of MSP1$_{FL}$-specific effector functions is a strong predictor of protection from malaria

Since we showed that each MSP1$_{FL}$-specific Fc-effector function was significantly associated with a reduced risk of requiring treatment upon sporozoite challenge, we next wanted to assess the contribution of the breadth of MSP1$_{FL}$-specific function to protective immunity. Therefore, we developed breadth scores of functional activities for every volunteer. We categorized the level of the five Fc-mediated functions for the study participants as either high or low (coded 1 or 0, respectively) based on function-specific thresholds (Nkumama et al, 2022 Preprint). We then summed the breadth scores across the five functions, such that individuals had a breadth score between 0 and 5. The proportion of individuals with a breadth score of 5 was significantly higher in the non-treated compared with the treated group (67/86, 78% versus 16/56, 29%, respectively, $P < 0.001$ [Fig 6A]). In a Kaplan-Meier survival analysis, the proportion of individuals who did not require treatment with a breadth score of 0 was 7%, compared with 81% for those with a breadth score of 5 ($P < 0.0001$, Fig 6B). Finally, we observed that the breadth of function increased with rising levels of anti-MSP1$_{FL}$ IgG (Fig 6C).

## Discussion

We found that anti-MSP1$_{FL}$ antibodies induced functional activity across multiple immune effectors. This agrees with our previous work assessing functional antibodies targeting merozoites (Nkumama et al, 2022 Preprint; Odera et al, 2023), and consistent with the fact that MSP1 is the most abundant protein on the surface of merozoites (Gilson et al, 2006). Individual Fc-mediated mechanisms have been correlated with protection from malaria in unrelated studies (Joos et al, 2010; Hill et al, 2013; Osier et al, 2014; Boyle et al, 2015; Tiendrebeogo et al, 2015; Murungi et al, 2016; Kana et al, 2019). Antibodies against MSP1$_{19}$ have been shown to induce phagocytosis and to activate complement via C1q (Kana et al, 2019; Reiling et al, 2019). We now show that MSP1 induces antibodies that also activate natural killer cell-mediated antibody-dependent

---

Fc-mediated natural killer cell degranulation; IFNγ, Fc-mediated natural killer IFNγ production; ADRB, antibody-dependent respiratory burst by neutrophils; OPA, opsonic phagocytosis activity of MSP1$_{FL}$-coupled microsphere beads by monocytes; AbC', antibody-dependent complement fixation activity.

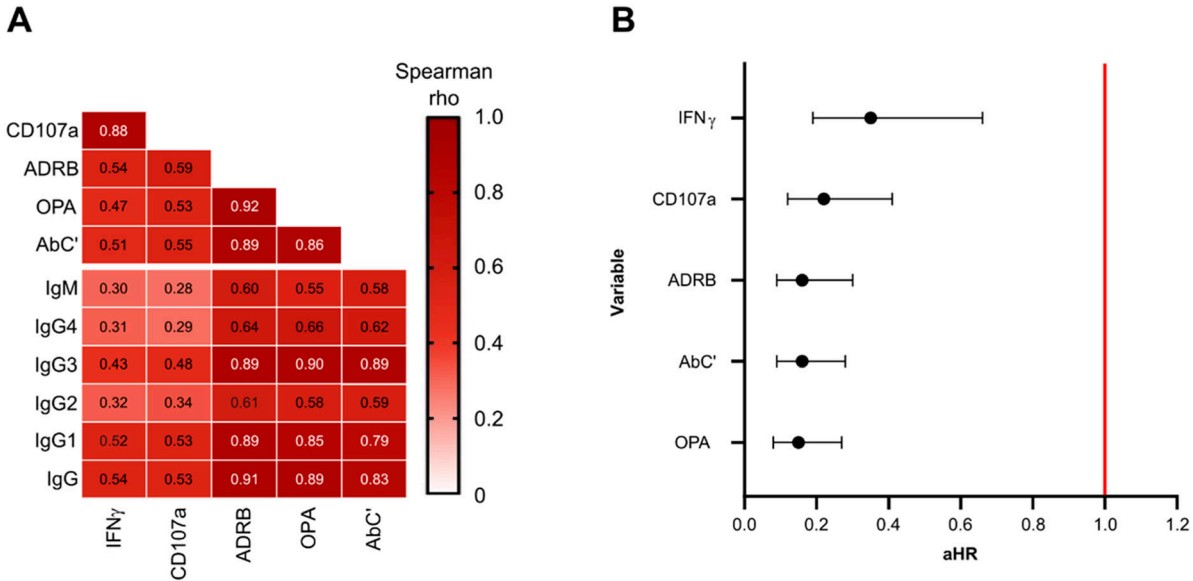

**Figure 5. Correlations between MSP1_{FL}-specific functional activities, isotype antibodies, and with protection.**
**(A)** A heatmap with correlation matrix showing spearman rank correlation coefficients for antibody levels and Fc-mediated effector functions. The color intensity represents the strength of correlation. **(B)** Forest plot showing adjusted hazard ratios (aHR) for MSP1_{FL}-specific functional activities ranked from lowest to highest. The aHRs were calculated using the cox regression model comparing the time to treatment between high versus low responders based on function-specific thresholds when adjusting for confounders (drug levels and year of study). Error bars indicate 95% confidence intervals and the red line indicates no protection (aHR = 1.0). CD107a, Fc-mediated natural killer cell degranulation; IFNγ, Fc-mediated natural killer IFNγ production; ADRB, antibody-dependent respiratory burst by neutrophils; OPA, opsonic phagocytosis activity of MSP1_{FL}-coupled microsphere beads by monocytes; AbC', antibody-dependent complement fixation activity.

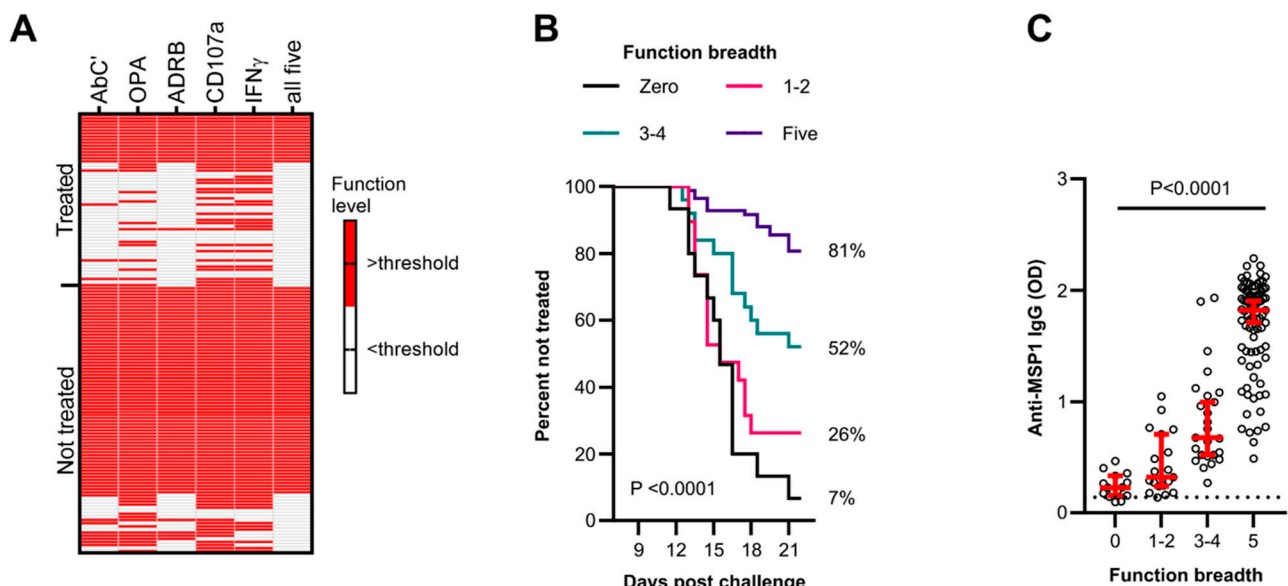

**Figure 6. Breadth of MSP1_{FL}-specific effector functions is a strong predictor of protection from malaria.**
**(A)** A heatmap showing the activity levels of all five Fc-mediated effector functions in treated (n = 56) and non-treated (n = 86) volunteers. Responses above a function-specific threshold are shown in red. Each column is a Fc-mediated function whereas each row is a single volunteer. AbC', antibody-dependent complement fixation activity; OPA, opsonic phagocytosis activity of MSP1_{FL}-coupled microsphere beads by monocytes; ADRB, antibody-dependent respiratory burst by neutrophils; IFNγ, Fc-mediated natural killer IFNγ production; CD107a, Fc-mediated natural killer cell degranulation. **(B)** Kaplan-Meier plot of volunteers who remained non-treated at different timepoints over the course of 21 d post challenge. Each line represents a function breadth score starting from 0 (n = 15), 1–2 (n = 19), 3–4 (n = 25), and 5 (n = 83). Significance was assessed using the Log rank sum test. **(C)** Anti-MSP1_{FL} IgG levels were compared between individuals with different levels of MSP1_{FL}-specific breadth of function. Each dot represents one sample in duplicates measured in one experiment. Error bars represent the median plus 95% confidence intervals. The seropositivity cut-off value was calculated as the optical density (OD) of malaria-naïve plasma samples plus three standard deviations indicated as the dotted line. Statistical differences were calculated using the Kruskal-Wallis test.

cellular cytotoxicity (Ab-NK) (Odera et al, 2023). In addition, the breadth of MSP1$_{FL}$-induced Fc-function across multiple immune effectors has not previously been evaluated in a human challenge study. As observed with IgG against merozoites (Nkumama et al, 2022 Preprint), the breadth of anti-MSP1$_{FL}$ IgG Fc-mediated function was more strongly associated with protection than individual mechanisms.

MSP1 has been extensively analysed in the context of naturally acquired immunity in traditional cohort studies and after vaccination. However, most studies focused on the conserved C-terminal subunit representing ~20% of the molecule (Malkin et al, 2007; Ogutu et al, 2009; Ellis et al, 2010; Fowkes et al, 2010; Richards et al, 2013). Although we found that immunogenic epitopes are distributed throughout MSP$_{FL}$, this is not new. Early studies showed that a variety of N-terminal subunits were immunogenic, in addition to the C-terminal (Chizzolini et al, 1988; Muller et al, 1989; Fruh et al, 1991; Tolle et al, 1993). This was replicated in later studies using the same subunits of MSP1 as those analysed in our study (Woehlbier et al, 2006), whereas other investigators focused on the highly polymorphic block 2 of the N-terminal (Cavanagh et al, 2004; Osier et al, 2008). Likewise, the finding the anti-MSP1 antibodies induce a range of functional activities is not new (Woehlbier et al, 2006; Jaschke et al, 2017; Rosenkranz et al, 2023). What is striking for studies of naturally acquired immunity is the strength of the association between this functional activity and protection from clinical episodes of malaria. We consider that this is in part because of the advantages of the human challenge platform that allowed greater precision in defining the exposure (infecting strain, dose, and timing), as well as the clinical outcome (twice daily sampling for malaria, in-residence monitoring for the onset of clinical symptoms, and exclusion of other infections) (Kapulu et al, 2018).

We found that anti-MSP1$_{FL}$ IgG antibodies were predominantly comprised of the cytophilic IgG1 and IgG3 isotypes. This is in keeping with data from multiple studies involving both the N- and C-termini of MSP1$_{FL}$ (Egan et al, 1995; Cavanagh et al, 2001; Cavanagh et al, 2004; Stanisic et al, 2009), with our own data using whole merozoites (Fig S2) and supports the well-established role for cytophilic antibodies in promoting immune effector activity that could be detrimental to parasites. Interestingly, a proportion of individuals had cytophilic anti-MSP1$_{FL}$ antibodies that induced Fc-mediated function across multiple effectors but were not protected from CHMI. An analysis of the correlation between these different effector functions revealed high correlations between complement activity, monocyte phagocytosis and neutrophil respiratory burst, but these were slightly lower for natural killer cell degranulation and IFNγ secretion. Taken together with the data showing that the breadth of effector function was the strongest correlate of protection, this may suggest that distinct epitopes within MSP1$_{FL}$ induce the full range of immune effectors. Similar observations have been made for the circumsporozoite surface protein, the major surface antigen of *P. falciparum* sporozoites. Whereas all regions of the protein were targets of opsonizing antibodies inducing phagocytosis of sporozoites, antibodies targeting the N-terminal domain in particular were most effective in activating phagocytes (Feng et al, 2021).

Alternatively, such antibodies may target similar epitopes but differ quantitatively or qualitatively. For example, the glycosylation

status of IgG antibodies has been shown to be critical for receptor engagement and induction of NK cell degranulation which played an important immunological role in malaria (Larsen et al, 2021) and other infectious diseases such as COVID-19 (Larsen et al, 2021). Additional possibilities are that FcγR polymorphisms account for the variation in protection despite the presence of anti-MSP1$_{FL}$ antibodies, or indeed that antibodies to other antigens are more important for a protective immune response in some individuals. Additional investigations in this regard will be necessary.

We were surprised to detect IgM antibodies against MSP1$_{FL}$ in malaria-exposed adults that presumably had life-long exposure to malaria parasites. Although we showed that antibody responses against MSP1$_{FL}$ were dominated by cytophilic IgG, recent findings have shown that IgM persists over time, activates complement-mediated invasion inhibition of red blood cells and is associated with protection from malaria (Boyle et al, 2019). In addition, it was demonstrated that IgM from malaria-exposed adults induces merozoite phagocytosis by monocytes (Hopp et al, 2021). However, research of FcμR distribution among innate immune cells such as macrophages has led to conflicting results (Uher et al, 1981; Kubagawa et al, 2009; Shima et al, 2010; Honjo et al, 2013; Lang et al, 2013; Hopp et al, 2021) in part because of differences in activation status of the cells and thus expression levels of FcμR. In this study we did not purify IgG or IgM because of limited sample volumes and thus could not assess its' specific functional activity. Nevertheless, we found that the correlations between IgM antibody levels and effector functions were relatively low. In addition, whereas the data on IgG and IgM cannot be directly compared because of differences in the quality of secondary reagents, it is tempting to speculate that the IgM findings were a red herring, as the antibody levels were relatively low compared with IgG. We cautiously speculate that MSP1$_{FL}$-specific IgM plays a minor role in protection from malaria amongst semi-immune adults.

IgG antibodies between the two main allelic variants of MSP1$_{FL}$ were highly correlated suggesting the presence of cross-reactive or shared epitopes and bodes well for vaccine development. Additional analyses using linear overlapping peptides confirmed not only that B-cell epitopes were present throughout the entire length of MSP1$_{FL}$, but also showed that in contrast to treated volunteers, IgG antibodies from those that did not require treatment preferentially targeted epitopes located within conserved and dimorphic regions. We did not have sufficient reagents to test whether these specific peptides induced functional antibodies in our panel of Fc-dependent assays. Importantly, these peptides are distributed across the entire MSP1 molecule and support our view that the full-length antigen should be prioritised for vaccine design. Indeed, a recent phase 1 clinical trial with full-length MSP1 (SumayaVac1) was safe and induced high titers of strain-transcending antibodies with complement fixation and ADRB activity in malaria-naïve adults (Blank et al, 2020).

We found that breadth of MSP1$_{FL}$-specific functional activity was low in the most treated volunteers; however, the fact that 29% of volunteers in whom the breadth of function was high were not protected from malaria was not surprising. Although MSP1$_{FL}$ is the most abundant antigen on the merozoite surface and therefore might dominate an antibody-mediated protective immune response against merozoites, other and/or multiple antigens could be important for

protection which has been highlighted in previous studies (Osier et al, 2008; Richards et al, 2013; Kana et al, 2019; Reiling et al, 2019). Moreover, we and others have recently shown that antibodies against antigens from other blood stages such as variant specific surface antigens on trophozoites (Kimingi et al, 2022) and ring-stage antigens (Musasia et al, 2022) were significantly associated with a reduced risk of developing clinical malaria in CHMI volunteers. As such, it remains plausible that antibodies targeting several blood stage antigens might work in synergy to confer protection. It is also possible that other antibody functions that were not measured as part of this investigation are important. These could include neutrophil-mediated phagocytosis (Garcia-Senosiain et al, 2021), antibody-dependent cellular inhibition by monocytes (ADCI) (Bouharoun-Tayoun et al, 1995), or ADCC by $\gamma\delta$ T cells (Farrington et al, 2020) amongst others.

MSP1 is cleaved before merozoite release and thought to be presented on the merozoite surface in complex with other merozoite antigens (Kadekoppala & Holder, 2010). Nevertheless, we were still able to detect functional antibodies against the full-length protein. Other investigators have detected functional antibodies against subunits of MSP1 (Kana et al, 2019; Reiling et al, 2019). Taken together, this suggests that the epitopes responsible for inducing the types of functional antibodies measured against $MSP1_{FL}$ are sufficiently exposed in vivo. Recently published work on vaccine-induced antibodies against $MSP1_{FL}$ in malaria-naïve individuals showed that functional antibodies were also induced against the subunits, particularly p83 and p42 (Rosenkranz et al, 2023). We cautiously speculate that the impact of the processing on functional immunity as measured in these studies is minimal.

A separate study using *E. coli* expressed antigens found functional antibodies primarily against p83 and p42, as well as the full-length antigen. This suggests that the antigens are all presented in a similar fashion during the assay (Rosenkranz et al, 2023). We are also unable to verify whether the data on linear B-cell epitopes are relevant to conformation sensitive epitopes (Kamuyu et al, 2018). Additional studies involving techniques such as electron based polyclonal epitope mapping (EMPEM) will be instructive in this regard (Han et al, 2021).

These limitations notwithstanding, we show that $MSP1_{FL}$-induced IgG antibodies that mediated Fc-functional activity across multiple immune effectors, mirroring what we had previously observed against merozoites in a controlled human challenge study. The strength of the association between $MSP1_{FL}$ functional activity and protection in individual assays, and collectively across multiple assays, support the conclusion that it is an important target of acquired immunity.

# Materials and Methods

### Study design and samples

The design of the Controlled Human Malaria Infection of Semi-Immune Kenyan Adults (CHMI-SIKA) study has been previously described (Kapulu et al, 2018). Briefly, 161 healthy Kenyan adults aged 18–45 yr with different antibody levels against schizont extract were recruited from three consecutive cohorts between 2016 and 2018. The volunteers were challenged with 3,200 cryopreserved *P. falciparum* NF54 sporozoites (Sanaria) by direct venous inoculation. Testing for blood stage parasitaemia was conducted by qRT-PCR twice per day between day 7 and 14 and once per day between day 15 and 21 post challenge. Volunteers were treated with artemether-lumefantrine when blood stage levels exceeded 500 parasites/$\mu$l, symptoms of clinical malaria with detectable parasites were recorded or after follow-up at day 22. Data from 19 volunteers were excluded because of antimalarial drug levels above the reported minimum inhibitory concentration in 12 volunteers and the presence of non-NF54 parasites in 7 volunteers. Thus, plasma samples from 142 volunteers and collected one day before challenge (C-1) were used in this study. 31% of the volunteers were female and the average age was 28.7 yr (range 18–45 yr).

### Ethics statement

The CHMI study was conducted at the KEMRI Wellcome Trust Research Programme in Kilifi, Kenya with ethical approval from the KEMRI Scientific and Ethics Review Unit (KEMRI//SERU/CGMR-C/029/3190) and the University of Oxford Tropical Research Ethics Committee (OxTREC 2–16). All participants gave written informed consent. The study was registered on ClinicalTrials.gov (NCT02739763), conducted based on good clinical practice, and under the principles of the Declaration of Helsinki.

### Expression of recombinant merozoite proteins

The Plasmid containing the codon-optimized sequence of full-length MSP1 (3D7) was received from the plasmid repository Addgene (#47709; Plasmid) which has been optimized for expression in mammalian cells (Crosnier et al, 2013; Kamuyu et al, 2018). $MSP1_{FL}$ was expressed in the Expi293 expression system (Gibco) following the manufacturer's instructions. In brief, $2 \times 10^6$ cells/ml of Exp1293F culture were transfected with the expression plasmid using the ExpiFectamine 293 transfection kit (Gibco). At 20 h post transfection, the transfection enhancers were added and after 5 d recombinant proteins were harvested and purified from culture supernatant using the Ni-NTA purification system (Invitrogen).

Recombinant MSP1 subunits and $MSP1_{FL}$-F were expressed in *E. coli* as previously described (Kauth et al, 2003; Kauth et al, 2006) and kindly provided by Sumaya Biotech.

### Indirect ELISA

Recombinant $MSP1_{FL}$ and MSP1 subunits were coated with 0.5 mg/well onto 96-well plates at 4°C overnight. The next day, the plates were washed with 1x PBS containing 0.05% Tween 20 (PBST) and blocked with 200 $\mu$l/well 1% skimmed milk for 2 h at RT. After blocking, the plates were washed followed by incubation with 50 $\mu$l/well serum samples at 1:1,000 for 2 h at RT. After incubation, plates were washed and 50 $\mu$l/well of respective secondary antibodies conjugated with HRP were added for 1 h at RT: rabbit anti-human IgG (Agilent), goat anti-human IgM (Thermo Fisher Scientific), and rabbit anti-human IgG1, 2, 3 or 4 (The Binding Site). Afterwards, the plates were washed and substrate solution (0.4 mg/ml O-phenylenediamine; Sigma-Aldrich) was added and incubated in the

dark at RT for 30 min. The reaction was stopped by adding 15 μl/well of 1 M hydrochloric acid (HCL) and the absorbance was read at 492 nm using the BioTek Cytation 3 cell imaging multi-mode reader and the Gen5 v3.02 software. The positive controls were a pool of hyperimmune serum from Kenyan adults (PHIS) (Murungi et al, 2013). The negative controls were plasma samples from malaria-naïve German adults and blank wells. All assays were conducted in duplicate and the final data presented for each individual represents the mean of duplicates. Samples were re-tested afresh and in duplicate if the coefficient of variation between duplicate results exceeded 20%. The cut-off for seropositivity was defined as the mean plus three SDs of the malaria-naïve negative controls and used to estimate the antibody prevalence.

### Mapping of linear B-cell epitopes

The mapping of linear IgG epitopes against full-length MSP1 was performed by PEPperPRINT GmbH, Heidelberg as previously described (Blank et al, 2020). Briefly, the sequence of full-length MSP1 (3D7) (UniProt ID: Pf3D7_0930300) was elongated with neutral GSGSGSG linkers at the N- and C-terminus to avoid truncated proteins. Next, the modified protein sequence was converted into short 15-mer amino acid peptides with a 14 amino acid overlap between the peptides. Subsequently, 1,720 unique peptides were printed in duplicates on the chip framed by additional HA (YPYDVPDYAG, 62 spots) and polio (KEVPALTAVETGAT, 62 spots) control peptides. The chip was blocked with Rockland blocking buffer MB-070 for 30 min, followed by pre-staining with secondary goat anti-human IgG (Fc) DyLight680 (0.1 μg/ml) and the monoclonal mouse anti-HA control antibody (12CA5) DyLight800 (0.5 μg/ml) in incubation buffer (washing buffer with 10% blocking buffer) for 45 min. Next, other copies of the MSP1$_{FL}$ microarray were incubated with 40 randomly selected human plasma samples (10 per clinical subgroup) from the CHMI-SIKA study (1:1,000 in incubation buffer) for 16 h at 4°C with gentle shaking on a plate shaker. After washing with PBST, fluorescently labelled secondary antibodies were added for 45 min at room temperature. The signals were detected using an InnoScan 710-IR Microarray Scanner at scanning gains of 50/50 (red/green). Spot intensity quantification was based on 16-bit grey scale tiff files with higher dynamic range than the 24-bit colourized tiff files. The analysis of the data was performed with PepSlide Analyzer. A maximum spot-to-spot deviation of 40% was tolerated; otherwise, the intensity value was put to zero. Positive and negative control samples were included as described for the ELISA assays above.

### Antibody-dependent complement fixation (AbC′) assay

AbC′ activity of MSP1$_{FL}$-specific antibodies was assessed in a modified ELISA measuring fixation of C1q, the first component of the classical complement pathway using a published protocol (Boyle et al, 2015; Reiling et al, 2019). Briefly, 96-well plates (Thermo Fisher Scientific) were coated with 0.5 mg/well of MSP1$_{FL}$ overnight at 4°C. The next day, plates were washed four times with PBST and blocked with 200 μl/well of 1% Casein/PBS at 37°C for 2 h. After blocking, the plates were washed and 50 μl/well of plasma samples diluted at 1:10 in PBS were added for 2 h at 37°C. After incubation, the plates

were washed and 40 μl of recombinant C1q (Abcam) at 10 mg/ml diluted in blocking buffer was added for 30 min at 37°C. Thereafter, the plate was washed and incubated with 50 μl/well of sheep anti-human C1q-HRP (Abcam) at a 1:100 dilution in blocking buffer for 1 h at 37°C. After washing, 50 μl/well of O-phenylenediamine solution (Sigma-Aldrich) was added and incubated for 45 min at room temperature before the reaction was stopped by adding 15 μl/well of 1 M HCL. The absorbance was read at 492 nm using the BioTek Cytation 3 cell imaging multi-mode reader and the Gen5 v3.02 software. Positive and negative control samples were included as described for the ELISA assays above.

### ADRB assay

The ADRB assay was performed as previously described (Kapelski et al, 2014; Nkumama et al, 2022 *Preprint*). Briefly, MSP1$_{FL}$ at 0.5 mg/well was coated onto opaque 96-well plates overnight (Greiner) at 4°C. The next day, the plates were washed with sterile PBS and blocked with 200 μl/well of sterile 1% casein/PBS for 1 h at 37°C. After blocking, the plates were washed and incubated with 50 μl/well of plasma samples diluted in PBS at 1:10 for 1 h at 37°C.

Neutrophils were prepared from whole blood collected in Heparin vacutainers. Blood was diluted 1:1 with HBSS (Thermo Fisher Scientific), carefully layered on top upon 7 ml of Histopaque-1077 (Sigma-Aldrich) and centrifuged at 600$g$ for 15 min. The pellet containing the neutrophils was resuspended in HBSS, mixed with 3% Dextran in a 1:2 ratio and incubated for 1 h at RT. Next, the supernatant was collected and centrifuged at 500$g$ for 7 min at 4°C. The pellet was then resuspended in ice cold 0.2% NaCl for 30 s to lyse contaminating RBC followed by adding an equal volume of ice cold 1.6% NaCl to stop lysis. Afterwards, the cells were centrifuged and the neutrophils were resuspended in neutrophil buffer (0.1% BSA, 1% D-Glucose in HBSS) and counted using a hemocytometer. The concentration of viable neutrophils was adjusted to 10 × 10$^6$ cells/ml and the cells were kept on ice.

Next, the plates were washed and 50 μl/well of luminol (Sigma-Aldrich) at 0.04 mg/ml was added before adding 50 μl/well of neutrophils. Chemiluminescence at 450 nm was immediately read for every 2 min over a duration of 1.5 h using the Biotek Synergy 4 plate reader and the Gen 5 acquisition software. The maximal relative light unit (RLU) for each sample was generated and indexed based on responses of a pool of hyper immune sera from Kenyan adults (PHIS), the positive control. The ADRB index was calculated as: (RLU of samples)/(RLU of PHIS) × 100. Negative controls were also included as described for the ELISAs above.

### OPA assay

The opsonic phagocytosis assay of antigen-coupled beads was based on a published protocol (Kana et al, 2019). Briefly, polychromatic red 1 μm microsphere beads (Polysciences Inc.) were coupled with 30 mg of MSP1$_{FL}$ in borate buffer (Polysciences Inc.) overnight at room temperature in the dark when rotating. The next day, the beads were centrifuged, and supernatant was carefully removed. Next, the beads were blocked thrice in blocking buffer (10 mg/ml BSA in borate buffer) for 30 min when rotating and stored in PBS with 5% glycerol and 0.1% sodium azide at 4°C.

For opsonization, 50 $\mu$l containing $7.5 \times 10^6$ antigen-coupled beads were added to each well of 96-well U-bottomed plates followed by incubation with 50 $\mu$l/well of heat-inactivated serum samples diluted at 1:2,000 for 1 h at 37°C. After incubation, the plates were centrifuged at 2,000$g$ for 7 min and washed thrice with PBS. Beads were resuspended in 50 $\mu$l of THP1 cell culture medium (2 mM L-glutamine, 2 mM HEPES, 10% FCS, 1% penicillin-streptomycin in RPMI 1640 media) and 50,000 THP1 cells in 150 $\mu$l were added to each well for 30 min at 37°C. Phagocytosis was arrested by centrifugation at 1,200 rpm for 7 min at 4°C. Plates were washed with ice-cold FACS buffer (0.5% BSA and 2 mM EDTA in PBS) and subsequently fixed in 2% formaldehyde/PBS. Flow cytometry was used to quantify THP1 cells containing fluorescent beads in the PE channel on the FACS Canto II high-throughput system (BD biosciences). Data analysis was performed using FlowJo V10.

Phagocytosis activity for each sample was indexed against the positive control (PHIS). The phagocytosis index was calculated as (% of stained THP1 cells opsonized with samples)/(% of stained THP1 cells opsonized with PHIS) × 100. Negative controls were also included as described for the ELISAs above.

### Antibody-mediated natural killer cells activation (Ab-NK)

The Ab-NK assay was performed as previously described (Odera et al, 2023). In brief, 500 $\mu$g of MSP1$_{FL}$ was coated onto 96-well culture plates overnight at 4°C. The next day, the plates were washed with PBS and blocked for 1 h with 1% Casein/PBS at 37°C. After blocking, the plates were washed and incubated with 50 $\mu$l/well of serum samples (1:10) for 4 h at 37°C. NK cells were isolated from human blood samples from healthy malaria-naïve donors. First, PBMCs were harvested using density gradient separation, washed, and resuspended in NK cell culture medium (RPMI 1640 media with 2 mM L-glutamine supplemented with 10% FCS and 1% penicillin-streptomycin). NK cells were isolated from PBMCs by negative isolation using the NK cell isolation kit (Miltenyi Biotec) as per manufacturer's instructions. A mix containing $5 \times 10^5$ freshly isolated NK cells, anti-human CD107a PE (1:70; BD biosciences), brefeldin A (1:200; Sigma-Aldrich), and monensin (1:200; Sigma-Aldrich) was added into each well and incubated for 18 h at 37°C. After stimulation, NK cells were carefully transferred into 96-well V-bottomed plates, centrifuged at 1,500 rpm for 5 min at 4°C, and washed with ice-cold FACS buffer (1% BSA, 0.1% sodium azide in PBS). NK cell viability was assessed by staining with 10 $\mu$l/well of fixable viability dye eFluor520 (Thermo Fisher Scientific) for 10 min at 4°C. NK cell surface receptors were stained with 20 $\mu$l/well of an antibody mix consisting of anti-CD56 APC (1:17; BD biosciences) and anti-CD3 PE-Cy5 (1:33; BD biosciences) for 30 min at 4°C in the dark. Next, NK cells were washed and fixed in 80 $\mu$l/well of fixing solution (CellFIX; BD biosciences) for 10 min at 4°C and subsequently permeabilized in 80 $\mu$l/well of permeabilization buffer (permwash; BD biosciences) for 10 min at 4°C. Intracellular IFN$\gamma$ was detected by adding 30 $\mu$l/well of anti-IFN$\gamma$ PE-Cy7 (BD biosciences) diluted at 1:33 in permeabilization buffer for 1 h at 4°C in the dark. After intracellular staining, the cells were washed thrice with 150 $\mu$l/well permeabilization buffer and finally resuspended in 150 $\mu$l/well FACS buffer. Acquisition was performed on the FACS Canto II (BD biosciences) and the data were analysed using FloJo V10. NK cell activity (degranulation and IFN$\gamma$ expression) for each sample was indexed against the positive control (PHIS). The degranulation/IFN$\gamma$ index was calculated as (% of NK cell degranulation/IFN$\gamma$ of samples)/(% of NK cell degranulation/IFN$\gamma$ release of PHIS) × 100. Negative controls were also included as described for the ELISAs above.

### Statistical analysis

Data were analysed using Prism 9.3.1 (GraphPad), Stata (version 14) or R. The Mann-Whitney U test was used to compare medians between treated and non-treated groups. The Kruskal-Wallis test was used to compare the four phenotypes based on parasite growth densities (febrile, non-febrile, PCR+, and PCR−) combined by Dunn's test for multiple comparisons. Pairwise correlations were calculated using nonparametric Spearman's correlations. The Wilcoxon test combined with Hommel correction was used to identify epitopes that were significantly different between non-treated and treated volunteers. Functional activity levels were categorized into high versus low responses based on function-specific thresholds which were determined by using maximally selected rank statistics analysis method in R (Nkumama et al, 2022 Preprint). The categorized data were analysed to assess the association between breadth of functional activity and time to treatment using the Cox proportional hazards model. Potential confounders were fit to the model (cohort and antimalarial drug levels). The Log rank sum test was used to compare the Kaplan-Meier survival curves.

# Members of the CHMI-SIKA Study Team

Abdirahman I Abdi[3], Yonas Abebe[7], Philip Bejon[3,8], Peter F Billingsley[7], Peter C Bull[10], Zaydah de Laurent[3], Mainga Hamaluba[3], Stephen L Hoffman[7], Eric R James[7], Melissa C Kapulu[3], Silvia Kariuki[3], Domitila Kimani[3], Rinter Kimathi[3], Sam Kinyanjui[3,9,11], Cheryl Kivisi[11], Johnstone Makale[3], Kevin Marsh[3,8], Khadija Said Mohammed[3], Moses Mosobo[3], Janet Musembi[3], Jennifer Musyoki[3], Michelle Muthui[3], Jedidah Mwacharo[3], Kennedy Mwai[3,4], Francis Ndungu[3], Joyce M Ngoi[3], Patricia Njuguna[3], Irene N Nkumama[2,3], Omar Ngoto[3], Dennis O Odera[3], Bernhards Ogutu[9,12], Fredrick Olewe[9], Donwilliams Omuoyo[3], John Ong'echa[9], Faith HA Osier[3,6], Edward Otieno[3], Jimmy Shangala[3], Betty Kim Lee Sim[7], Thomas L Richie[7], James Tuju[3,5], Juliana Wambua[3], Thomas N Williams[3,13]

[7]Sanaria Inc., Rockville, MD, USA

[8]Centre for Tropical Medicine and Global Health, Nuffield Department of Medicine, University Oxford, Oxford, UK

[9]Centre for Clinical Research, Kenya Medical Research Institute, Kisumu, Kenya

[10]Department of Pathology, University of Cambridge, Cambridge, UK

[11]Pwani University, Kilifi, Kenya

[12]Center for Research in Therapeutic Sciences, Strathmore University, Nairobi, Kenya

[13]Department of Medicine, Imperial College London, London, UK

## Data Availability

Primary research data, statistical analyses applied, and methodologies are provided in the study results, figure legends, and uploaded supplemental materials. All raw data are available to share with the scientific community upon request.

## Supplementary Information

## Acknowledgements

We are grateful to all the study volunteers who have participated in the CHMI-SIKA study. We are also very grateful to the study teams at the study sites in Kilifi and Ahero, the collaborating teams at Sanaria, the study investigators, and all the clinical and laboratory teams. The CHMI-SIKA study was supported by a Wellcome Trust grant (107499) and sponsored by the University of Oxford. This work was supported in part by a Sofja Kovalevskaja Award from the Alexander von Humboldt Foundation (3.2 – 1184811-KEN-SKP), an EDCTP Senior Fellowship (TMA 2015 SF1001) which is part of the EDCTP2 programme supported by the European Union, and a Wellcome Discovery Award (226669/Z/22/Z) all awarded to FHA Osier. K Mwai and R Ogwang were supported by an NIHR Global Health Research Unit grant number 16/136/33; Tackling Infections to Benefit Africa (TIBA). K Mwai and R Ogwang were also supported through the DELTAS Africa Initiative Grant No.107754/Z/15/Z-DELTAS Africa SSACAB and from DELTAS Africa Initiative (DEL-15-003). The DELTAS Africa Initiative is an independent funding scheme of the African Academy of Sciences (AAS)'s Alliance for Accelerating Excellence in Science in Africa (AESA) and supported by the New Partnership for Africa's Development Planning and Coordinating Agency (NEPAD Agency) with funding from the Wellcome Trust (107769/Z/10/Z) and the UK government. The views expressed in this publication are those of the author(s) and not necessarily those of AAS, NEPAD Agency, Wellcome Trust or the UK government. We also thank Prof. Michael Lanzer and Sumaya Biotech for providing recombinant antigens.

### Author Contributions

M Rosenkranz: conceptualization, data curation, formal analysis, investigation, methodology, and writing—original draft, review, and editing.
IN Nkumama: formal analysis and writing—review and editing.
R Ogwang: investigation and writing—review and editing.
S Kraker: investigation and writing—review and editing.
M Blickling: formal analysis, investigation, and writing—review and editing.
K Mwai: formal analysis, methodology, and writing—review and editing.
D Odera: resources, methodology, and writing—review and editing.
J Tuju: resources, methodology, and writing—review and editing.
K Fürle: resources, methodology, project administration, and writing—review and editing.
R Frank: resources, methodology, project administration, and writing—review and editing.
E Chepsat: resources, methodology, and writing—review and editing.
MC Kapulu: conceptualization, resources, supervision, project administration, and writing—original draft, review, and editing.
CHMI-SIKA Study team: conceptualization, resources, data curation, supervision, funding acquisition, investigation, and writing—original draft and project administration.
FHA Osier: conceptualization, resources, data curation, supervision, funding acquisition, investigation, and writing—original draft and project administration.

### Conflict of Interest Statement

In the CHMI-SIKA team, Y Abebe, PF Billingsley, SL Hoffman, ER James, BK Lee Sim, and TL Richie are salaried, full-time employees of Sanaria Inc., the manufacturer of Sanaria PfSPZ Challenge. Thus, all authors associated with Sanaria Inc. have potential conflicts of interest. All other authors declare no conflict of interests.

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
