## [Reviewer comments · Life Science Alliance]

Life Science Alliance

Full-length MSP1 is a major target of protective immunity after controlled human malaria infection

Micha Rosenkranz, Irene Nkumama, Rodney Ogwang, Sara Kraker, Marie Blickling, Kennedy Mwai, Dennis Odera, James Tuju, Kristin Fürle, Roland Frank, Emily Chepsat, Melissa Kapulu, CHMI-SIKA Study team, and Faith Osier

DOI: <https://doi.org/10.26508/lsa.202301910>

Corresponding author(s): Faith Osier, Imperial College London and Micha Rosenkranz, University Hospital Heidelberg

Review Timeline:

Submission Date:	2023-01-08
Editorial Decision:	2023-02-13
Revision Received:	2024-03-05
Editorial Decision:	2024-03-20
Revision Received:	2024-05-08
Accepted:	2024-05-08

Transaction Report:

February 13, 2023

Re: Life Science Alliance manuscript #LSA-2023-01910

Prof. Faith Hope Among'in Osier
Imperial College London
Department of Life Science
Sir Alexander Fleming Building
Exhibition Road, South Kensington
London SW7 2AZ
United Kingdom

Dear Dr. Osier,

Thank you for submitting your manuscript entitled "Full-length MSP1 is a major target of protective immunity after controlled human malaria infection" to Life Science Alliance. The manuscript was assessed by expert reviewers, whose comments are appended to this letter. We invite you to submit a revised manuscript addressing the Reviewer comments.

Thank you for this interesting contribution to Life Science Alliance. We are looking forward to receiving your revised manuscript.

Sincerely,

B. MANUSCRIPT ORGANIZATION AND FORMATTING:

Reviewer #1 (Comments to the Authors (Required)):

The manuscript by Osier et al. describes the correlation between pre-challenge levels of antibodies against the full-length Plasmodium falciparum protein, MSP1, and the antibodies respective Fc-mediated functional activity, with protection against malaria in semi-immune volunteers. Overall, the manuscript is well written, and the data supports the authors claim that increased levels of anti-MSP1 full length antibodies, specifically total IgG, and subclasses IgG1 and IgG3, are correlated with increased protection against developing clinical malaria symptoms and high levels of parasitaemia. Additionally, the authors data support their findings that MSP1FL-specific antibodies from non-treated volunteers show an increase in Fc-mediated functional activity, demonstrated by multiple functional assays. The correlation analyses further emphasize the relationship between increased levels and breadth of Fc-mediated effector activity with a decrease in volunteers who required treatment. The authors are thorough in their investigations, demonstrating that the correlation between anti- MSP1FL antibodies and necessity for malaria treatment is consistent for both the two dominant allelic forms of MSP1, as well as against the different subunits of MSP1.

The authors should clearly state the definition of the volunteer groups (treated, untreated, febrile, and non-febrile) as it seems like all volunteers are treated post-day 22. In the manuscript, it would be helpful to include the final numbers of each group, including febrile, non-febrile, PCR+ and PCR- sub-groups, in a table.

Reviewer #2 (Comments to the Authors (Required)):

The need for a means to protect humans from malaria is profound, and there is much effort being expended in the identification of immunoprotective antigens to use in vaccine development. One, the MSP1 antigen, has been heavily studied, primarily as fragments. In this manuscript, the authors look at the immune response to full-length MSP1. This study piggybacks onto a controlled human malaria infection study (CHMI-SIKA), making use of sera from individuals of known malarial immunity status, and finds strong associations of various antibody-mediated mechanisms to immune protection, in the context of full-length MSP1. The manuscript is overall well-written, the Title and Abstract accurately reflect content, and the figures are of a high quality. This is a high-quality manuscript and few specific issues were identified in this review. Its primary shortcomings are possible overinterpretation and extrapolation, particularly concerning mechanisms, and what feels like promotion of the full-length protein as a potential vaccine antigen rather than a dispassionate assessment of it as such. Issues that would benefit this manuscript if corrected include the below.

1. The authors nicely demonstrate associations of antibody responses to epitopes within the full-length protein with protection. Many of the recognized epitopes are ones found within regions of the protein studied previously as fragments, but the authors do not provide recognition of that fact. Rather, they treat the full-length protein as though it were somehow unrelated to the materials used in other studies.
2. It would be helpful if the authors were to give a little contextual information when they start describing the results. Although I could guess, until I looked up the CHMI-SIKA study itself it was not clear to me what was meant by "treated" and "non-treated" volunteers (for example, drug-treated? immunized?). It would be appropriate to give context here and it will clarify what follows.
3. Figures S1 and S2 are mislabeled as Figures S2 and S3.
4. In presenting the mapping of linear epitopes by the peptide chip approach a number of epitopes are listed, yet almost none seem to have elicited responses. The epitopes of the responses observed are not identified. It would be helpful to do so. Also, it would be helpful visually if the protein schematic were at the top of the figure, immediately above the response lines.
5. In describing the breadth of responses, the authors state that, "...Fc-effector function predicted significantly reduced risk..". This is an overinterpretation. In fact, the authors showed strong statistical associations of those functions with protection status. The responses to MSP1 could be merely coincidental to immune responses mounted to other parasite components. In describing "breadth scores" values of 0 or 1 were assigned, but the outcomes are presented in percentages. Please explain the relationship between breadth scores and percentages.
6. This study did not include any experimental results allowing direct comparison of the degree of association with immune protection of the various Fc-mediated mechanisms with that of invasion inhibition, the assay used in most studies involving MSP1. This makes it inherently difficult to argue superiority of one antigen form versus another, reducing the impact of this manuscript. If such data were generated they need to be included.

7. The mapping of linear epitopes is quite interesting, but it is not at all clear how these relate to the conformational epitopes that the individual's antibodies would encounter in an actual infection. The authors allude to this problem once in the Discussion but ignore this issue throughout the remainder of the manuscript. Even the accessibility of these linear epitopes on the native protein is unclear, and unless accessible on a live parasite are unlikely to be protective against merozoite stages. The structure of MSP1, a dimer, is available (Dijkman, P.M. et al. 2021. *Sci. Adv.* 7(23): abg0465). The authors are encouraged to provide information on how the potentially protective linear epitopes map onto the actual structure.

8. Throughout the manuscript the grouped sera provide for comparisons, with the most severely unprotected serving essentially as negative biological controls for strongly protected individuals. True negative sera from known malaria-naïve individuals should have been included. Similarly, no mention is made of true negative control antigens in any of the assays. The use of maximally selected rank statistics is an accepted statistical approach to establish test thresholds that will segregate outcomes into two groups. However, in this case this includes the assumption that the treated and non-treated groups should be segregated based upon the parameter being measured (assay reactivity dependent upon MSP1fl). This creates mathematically-supported circular reasoning that depends upon the hypothesis it's intended to test. These data would be more convincing if supported throughout by true assay controls. Please acknowledge this weakness in the Discussion.

9. Figure 1 legend. It would be helpful to simply state these data are the results of an antibody-capture ELISA test, so that the reader doesn't have to search the Methods section to find out what was done.

Reviewer #3 (Comments to the Authors (Required)):

Rosenkranz et al have utilised serum from a cohort of malaria exposed adults living in Kenya to determine prevalence and subtypes of anti-MSP1 antibodies. They found that individuals who had better immunity to parasite challenge (defined by an individual requiring no treatment during the 21-day trial) had higher pre-trial levels of anti-MSP1 antibodies with broad effector mechanisms. The manuscript is mostly well written and the data is well put together. All experiments have only been done once, which inevitably raises reproducibility concerns, but this is somewhat mitigated by the number of samples tested across the study and the consistency of the trends. While the authors demonstrate that these individuals develop antibodies to epitopes which span the full-length protein, it was not possible to ascertain whether the repertoire of functional antibodies are directed toward particular epitopes of MSP1. The inability to demonstrate this weakens the overall conclusion that full length MSP1 is a significantly superior target to specific regions of the protein.

Major comments

- The authors have not detailed the sequence of proteolytic cleavage for MSP1 by SUB1 and SUB2 during egress and prior to host cell attachment in the manuscript. Although a visual representation is provided in Fig 3, it is not really highlighted as a means to understand the cleavage pattern/subunits themselves and knowledge of this process is assumed. It then becomes a challenge to keep track of what and where the subunits are as they are used throughout the results text, and this would particularly be the case for those who are not experts on the protein. I strongly suggest that a short passage be added to the introduction to outline the steps of cleavage and the subsequent subunits.

- The MSP1 protein is cleaved prior to merozoite release and the protein presented on the merozoite surface is thought to be a processed antigen in complex with other antigens. Could the authors please consider and discuss the implications of the processed antigen as a target versus the whole full length MSP1 used in the majority of assays.

- It is not entirely clear how participants were classified as treated febrile/non-febrile (was this done by blood smear microscopic examination?). Similarly, how PCR+ve/-ve groups were separated was not explained. A clear explanation of how participants were classified as one of these four groups from the original population would be really useful. This should go in at the start of the results since this terminology is used immediately with no background as to how the classification works.

Similarly, in the Figure legends, the term parasite growth patterns is used in the Figure legends, but nowhere else. It would be very beneficial to the manuscript to explain why the groups were partitioned as they were and use similar nomenclature throughout.

- The authors have concluded from the data in this paper that full length MSP1 is a superior antigen for measuring correlates of protection. However, a full comparison with functional antibody responses to subunits using these samples was not undertaken, which weakens this conclusion. I suggest that the limitations of the comparison made in this regard (mentioned in 2nd last paragraph: 'We were unable to assess the functional activity of antibodies directed against individual subunits and specific peptides within MSP1') and outstanding reasons that still support this claim, should be discussed and compared directly in more detail since this is a key premise for the conclusion that full length MSP1 is a superior correlate of immunity of subunits. Have the authors considered whether the way full length protein is presented in these assays differs from that of the subunits and whether this could also have a positive impact on the immune measures?

Minor comments

- First section of results "anti-MSP1FL antibody levels were higher in non-treated versus treated CHMI-SIKA volunteers" the last paragraph refers to Figure S2, this needs to be changed to Figure S3.

- Introduction:

'In other studies, antibodies targeting MSP1 have been shown to promote several mechanisms...'

Suggest change to

'In other studies, antibodies targeting regions of MSP1 have been shown to promote several mechanisms....'.

-Results:

'We categorized the level of Fc-mediated functions of the study participants as either high or low (coded 1 or 0, respectively) based on function specific thresholds. We then summed the breadth scores across the five functions.'

Suggest changing to.

'We categorized the level of the 5 Fc-mediated functions for the study participants as either high or low (coded 1 or 0, respectively) based on function specific thresholds. We then summed the breadth scores across the five functions.'

-By convention, all measurement values should have a space before the SI units.

Dear Editors and Reviewers

Thank you for the opportunity to respond to the reviewers comments. We believe the manuscript is greatly improved as a result. We have taken on board the sentiments of Reviewers #2 and #3 to the effect that we did not demonstrate that full-length MSP1 is superior to the subunits. To this end, we have removed all reference to this in our discussion and conclusion and restricted ourselves to what the data show. We have also removed statements that were promoting MSP1 as a vaccine candidate. We have updated referencing throughout, and additional major corrections in this regard are highlighted immediately below. We have introduced line numbers to facilitate locating the changes and switched off track changes.

We previously referred to the preprints of two papers. One of these was published in 2023 and has been updated throughout (Odera 2023, *Sci Transl Med*). We have just been informed that the second one will be accepted shortly, we hope to update that with the published version once this review is complete (Nkumama, *bioRxiv*). Both these references appear twice in the current document due to formatting constraints which can hopefully be resolved once the reviewers are happy with the revisions and the paper is accepted.

Major changes are highlighted in blue in the submission.

Introduction

Paragraph 1 (Lines 32 - 41) – We updated the WHO world malaria report and refreshed the section to reflect the fact that there are now 2 malaria vaccines licensed.

Paragraph 3 (Lines 63 – 68) – Introduced new text to discuss the processing of MSP1 see Reviewer #3

Paragraph 5 (Lines 86 – 99) – Introduced new text refocusing on a previously undiscussed strength of the study which is the utilisation of the human challenge model for the work

Paragraph 6 (Lines 102 – 103) – Introduced the human challenge study called CHMI-SIKA

Results

CHMI-SIKA Study outcome (Lines 109 – 120) – provided the details of the study, the definitions of the volunteer subgroups in an introductory descriptive section – All reviewers

Discussion

Paragraph 1 (Lines 253 – 266) – Reframed the discussion away from comparisons of subunits

Paragraph 2 (Lines 267 – 284) – Discussed older data – clarifying what is new

Paragraph 8 (Lines 355 – 364) – Discussed MSP1 processing and likely impact on immune response

Paragraphs 9 & 10 (Lines 365 – 376) – Reframed the limitations and concluding remarks in response to the reviewers as discussed above.

Point by point responses to the reviewers are provided in blue below.

Reviewer #1 (Comments to the Authors (Required)):

The manuscript by Osier et al. describes the correlation between pre-challenge levels of antibodies against the full-length Plasmodium falciparum protein, MSP1, and the antibodies respective Fc-mediated functional activity, with protection against malaria in semi-immune volunteers. Overall, the manuscript is well written, and the data supports the authors claim that increased levels of anti-MSP1 full length antibodies, specifically total IgG, and subclasses IgG1 and IgG3, are correlated with increased protection against developing clinical malaria symptoms and high levels of parasitaemia. Additionally, the authors data support their findings that MSP1FL-specific antibodies from non-treated volunteers show an increase in Fc-mediated functional activity, demonstrated by multiple functional assays. The correlation analyses further emphasize the relationship between increased levels and breadth of Fc-mediated effector activity with a decrease in volunteers who required treatment. The authors are thorough in their investigations, demonstrating that the correlation between anti- MSP1FL antibodies and necessity for malaria treatment is consistent for both the two dominant allelic forms of MSP1, as well as against the different subunits of MSP1.

The authors should clearly state the definition of the volunteer groups (treated, untreated, febrile, and non-febrile) as it seems like all volunteers are treated post-day 22. In the manuscript, it would be helpful to include the final numbers of each group, including febrile, non-febrile, PCR+ and PCR- sub-groups, in a table.

We thank the reviewer for these positive comments. The data the reviewer requests have been introduced as a paragraph describing the outcome of the study at the beginning of the results section. This provides both the definitions of the volunteer groups, as well as the final numbers in each group.

CHMI-SIKA Study Outcome (Lines 109 – 120)

“Of the 142 study participants who were included in the final analysis, a proportion were treated before the end of the study on day 22 (39%, 56/142) while the remainder were not (61%, 86/142), referred to as “treated” and “non-treated”, respectively. Treatment before the end of the study was provided if the volunteers developed clinical symptoms of malaria, including a fever (subclassified as febrile, n = 26) or if there were no clinical symptoms but parasitaemia exceeded a predefined threshold of 500 parasites/ μ l (subclassified as non-febrile). Artemether-Lumefantrine was used for treatment. Volunteers not requiring treatment before the end of the study were subclassified into those that were PCR positive for blood stage malaria parasites (n = 53) and those that were PCR negative (n = 33). At the end of the study and before discharge on day 22, all volunteers that remained free of clinical symptoms were treated to ensure that all infections were cleared.”

Reviewer #2 (Comments to the Authors (Required)):

The need for a means to protect humans from malaria is profound, and there is much effort being expended in the identification of immunoprotective antigens to use in vaccine development. One, the MSP1 antigen, has been heavily studied, primarily as fragments. In this manuscript, the authors look at the immune response to full-length MSP1. This study piggybacks onto a controlled human malaria infection study (CHMI-SIKA), making use of sera from individuals of known malarial immunity status, and finds strong associations of various antibody-mediated mechanisms to immune protection, in the context of full-length MSP1. The manuscript is overall well-written, the Title and Abstract accurately reflect content, and the figures are of a high quality. This is a high-quality

manuscript and few specific issues were identified in this review. Its primary shortcomings are possible overinterpretation and extrapolation, particularly concerning mechanisms, and what feels like promotion of the full-length protein as a potential vaccine antigen rather than a dispassionate assessment of it as such. Issues that would benefit this manuscript if corrected include the below.

1. The authors nicely demonstrate associations of antibody responses to epitopes within the full-length protein with protection. Many of the recognized epitopes are ones found within regions of the protein studied previously as fragments, but the authors do not provide recognition of that fact. Rather, they treat the full-length protein as though it were somehow unrelated to the materials used in other studies.

We agree with the reviewers judgement and have acknowledged previous work early in the discussion, including early studies conducted before the year 2000, and more recent studies. This covers studies that analyzed immunogenicity as well as antibody function. We have also removed statements suggesting promote MSP1 as a vaccine candidate throughout the discussion and simply presented the facts to avoid overinterpretation. The paragraph below is now included in the discussion (Lines 267 – 284)

MSP1 has been extensively analysed in the context of naturally-acquired immunity in traditional cohort studies and following vaccination. However, most studies focused on the conserved C-terminal subunit representing ~20 % of the molecule (Ellis et al., 2010; Fowkes et al., 2010; Malkin et al., 2007; Ogutu et al., 2009; Richards et al., 2013). Although we found that immunogenic epitopes are distributed throughout MSP_{FL}, this is not new. Early studies showed that a variety of N-terminal subunits were immunogenic, in addition to the C-terminal (Chizzolini et al., 1988; Früh et al., 1991; Muller et al., 1989; Tolle et al., 1993). This was replicated in later studies using the same subunits of MSP1 as those analyzed in our study (Woehlbier et al., 2006), while other investigators focused on the highly polymorphic block 2 of the N-terminal (Cavanagh et al., 2004; Osier et al., 2008). Likewise, the finding the anti-MSP1 antibodies induce a range of functional activities is not new (Jäschke et al., 2017; Rosenkranz et al., 2023; Woehlbier et al., 2006)). What is striking for studies of naturally acquired immunity is the strength of the association between this functional activity and protection from clinical episodes of malaria. We consider that this is in part due to the advantages of the human challenge platform, that allowed greater precision in defining the exposure (infecting strain, dose and timing), as well as the clinical outcome (twice daily sampling for malaria, in-residence monitoring for the onset of clinical symptoms, and exclusion of other infections) (Kapulu et al., 2018).

2. It would be helpful if the authors were to give a little contextual information when they start describing the results. Although I could guess, until I looked up the CHMI-SIKA study itself it was not clear to me what was meant by "treated" and "non-treated" volunteers (for example, drug-treated? immunized?). It would be appropriate to give context here and it will clarify what follows.

3. Figures S1 and S2 are mislabeled as Figures S2 and S3.

This has been clarified in response to Reviewer #1 above (Lines 109 – 120). Figures S1 and S1 are now correctly labelled.

4. In presenting the mapping of linear epitopes by the peptide chip approach a number of epitopes are listed, yet almost none seem to have elicited responses. The epitopes of the responses observed are not identified. It would be helpful to do so. Also, it would be helpful visually if the protein schematic were at the top of the figure, immediately above the response lines.

We respectfully refer the reviewer to the results section for this data (**Figure 3**) where we defined the positions of the most dominant epitopes using the amino acid positions, and indicating the fragment in which the epitope was located. **Lines 185 to 188** have the following statement “Dominant epitopes were found in the conserved N- and C-terminal regions (EEITTK, position 56-61, p83 and SPLKTLSEVSIQTE, position 1150-1163, p38) and central dimorphic domains (ETEETEET, position 747-754, p30).”

We believe that these are difficult for the reviewer to visualize because the lowest two categories on the figure are derived from non-responders i.e. the treated groups comprising febrile and non-febrile subgroups. The dominant responses are clearly visible in the top 2 panels representing the PCR+ and PCR- subgroups. We regret that moving the protein schematic to the top is not feasible as it makes it very difficult to show the amino acids which are of varying length. We recommend that the reviewer also refers to Supplementary Table 1 which provides additional detail for peptides where the responses differed significantly between groups.

Antibodies from most of the volunteers that were treated (both febrile and non-febrile) bound to just a few epitopes with a lower magnitude of response. In contrast, antibodies from the non-treated PCR+ and PCR- volunteers bound to more epitopes, with a higher magnitude of response. The strongest responses were observed in conserved and dimorphic regions of MSP1

5. In describing the breadth of responses, the authors state that, “..Fc-effector function predicted significantly reduced risk..”. This is an overinterpretation. In fact, the authors showed strong statistical associations of those functions with protection status. The responses to MSP1 could be merely coincidental to immune responses mounted to other parasite components. In describing “breadth scores” values of 0 or 1 were assigned, but the outcomes are presented in percentages. Please explain the relationship between breadth scores and percentages.

We agree with the reviewer and have restricted the description to “association” as opposed to “prediction” at the beginning of the last results section and throughout the manuscript.

We have clarified the relationship between the scores and percentages as described below (Lines 242 – 248).

“We categorized the level of the 5 Fc-mediated functions for the study participants as either high or low (coded 1 or 0, respectively) based on function specific thresholds. (Nkumama et al., 2022 *Preprint*). We then summed the breadth scores across the five functions, such that individuals had a breadth score between 0 and 5. The proportion of individuals with a breadth score of 5 was significantly higher in the non-treated compared to the treated group (67/86, 78% versus 16/56, 29%, respectively, $p < 0.001$, **Fig 6A**).”

6. This study did not include any experimental results allowing direct comparison of the degree of association with immune protection of the various Fc-mediated mechanisms with that of invasion inhibition, the assay used in most studies involving MSP1. This makes it inherently difficult to argue superiority of one antigen form versus another, reducing the impact of this manuscript. If such data were generated they need to be included.

We agree with the reviewer that these data are not shown specifically for MSP1. Our sample volumes did not allow us to affinity-purify enough MSP1-specific antibodies to demonstrate this. We therefore remove the assertion that MSP1 activity in the Fc-dependent assays is superior to its activity in invasion-inhibition assays, which was the argument presented for example in the opening

paragraph of the discussion. We have re-organised the discussion to focus on the main finding as presented in the introductory comments.

7. The mapping of linear epitopes is quite interesting, but it is not at all clear how these relate to the conformational epitopes that the individual's antibodies would encounter in an actual infection. The authors allude to this problem once in the Discussion but ignore this issue throughout the remainder of the manuscript. Even the accessibility of these linear epitopes on the native protein is unclear, and unless accessible on a live parasite are unlikely to be protective against merozoite stages. The structure of MSP1, a dimer, is available (Dijkman, P.M. et al. 2021. Sci. Adv. 7(23): abg0465). The authors are encouraged to provide information on how the potentially protective linear epitopes map onto the actual structure.

We agree entirely with the reviewer for these discerning comments. Reactivity to linear epitopes does not automatically translate to reactivity to conformational epitopes. However, the fact that we detected strong antibody responses to a handful of linear epitopes that were differentially recognized between treated and non-treated individuals suggests these ones are accessible on live parasites. In the absence of similar mapping studies on the full-length antigen for example using electron microscopy polyclonal epitope mapping (EMPEM), any statements we may make will be purely speculative. Additional studies are planned in collaboration with a structural biologist, and we hope to shed more light on this in the future. We refer to this in the discussion (Lines 367 – 370).

8. Throughout the manuscript the grouped sera provide for comparisons, with the most severely unprotected serving essentially as negative biological controls for strongly protected individuals. True negative sera from known malaria-naïve individuals should have been included. Similarly, no mention is made of true negative control antigens in any of the assays. The use of maximally selected rank statistics is an accepted statistical approach to establish test thresholds that will segregate outcomes into two groups. However, in this case this includes the assumption that the treated and non-treated groups should be segregated based upon the parameter being measured (assay reactivity dependent upon MSP1fl). This creates mathematically-supported circular reasoning that depends upon the hypothesis it's intended to test. These data would be more convincing if supported throughout by true assay controls. Please acknowledge this weakness in the Discussion.

We regret that we overlooked the mention of controls. Each experiment included positive, negative and unopsonized controls. Positive controls comprised a serum pool from malaria-exposed adults living in Kenya with known reactivity against merozoites and MSP1 (Pooled HyperImmune Serum, PHIS). Negative controls were serum samples from German adults with no prior exposure to malaria. For the unopsonized control, no serum was added to antigen coated plates or beads to test for any non-specific Fc-mediated effector activity. We've added this information to the relevant methods sections (Lines 424 – 430; 452; 467; 491; 513 – 514 and 544)

9. Figure 1 legend. It would be helpful to simply state these data are the results of an antibody-capture ELISA test, so that the reader doesn't have to search the Methods section to find out what was done.

We have added the reference to the ELISA in the title of the legend to Figure 1 (Line 982)

Reviewer #3 (Comments to the Authors (Required)):

Rosenkranz et al have utilised serum from a cohort of malaria exposed adults living in Kenya to

determine prevalence and subtypes of anti-MSP1 antibodies. They found that individuals who had better immunity to parasite challenge (defined by an individual requiring no treatment during the 21-day trial) had higher pre-trial levels of anti-MSP1 antibodies with broad effector mechanisms. The manuscript is mostly well written and the data is well put together. All experiments have only been done once, which inevitably raises reproducibility concerns, but this is somewhat mitigated by the number of samples tested across the study and the consistency of the trends. While the authors demonstrate that these individuals develop antibodies to epitopes which span the full-length protein, it was not possible to ascertain whether the repertoire of functional antibodies are directed toward particular epitopes of MSP1. The inability to demonstrate this weakens the overall conclusion that full length MSP1 is a significantly superior target to specific regions of the protein.

Major comments

- The authors have not detailed the sequence of proteolytic cleavage for MSP1 by SUB1 and SUB2 during egress and prior to host cell attachment in the manuscript. Although a visual representation is provided in Fig 3, it is not really highlighted as a means to understand the cleavage pattern/subunits themselves and knowledge of this process is assumed. It then becomes a challenge to keep track of what and where the subunits are as they are used throughout the results text, and this would particularly be the case for those who are not experts on the protein. I strongly suggest that a short passage be added to the introduction to outline the steps of cleavage and the subsequent subunits.

A paragraph describing the processing and subunits has been added to the introduction (Lines 63 – 68).

“MSP1 is expressed as a ~190-kDa precursor protein that is enzymatically processed by subtilisin-like protease 1 (SUB-1) generating 4 major subunits, p83, p30, p38 and p42. The subunits remain non-covalently attached to each other and are tethered to the plasma membrane via a glycosylphosphatidylinositol (GPI) anchor. During merozoite invasion, SUB-2 cleaves C-terminal p42 resulting in the formation of p33 and p19; the latter gets internalized during invasion while the rest of the protein is shed (Blackman 1992).”

The MSP1 protein is cleaved prior to merozoite release and the protein presented on the merozoite surface is thought to be a processed antigen in complex with other antigens. Could the authors please consider and discuss the implications of the processed antigen as a target versus the whole full length MSP1 used in the majority of assays.

We have attempted to address this point in the discussion, referring to a recent publication where functional responses to the fragments were analyzed, albeit in a malaria-naive population, and using vaccine-induced responses. We only had sufficient antigen for ELISA assays in our own investigation. Based on these data, and those from other investigators who have examined Fc IgG-mediated responses to MSP1, we cautiously speculate that the impact of antigen processing on the generation of functional immune responses is minimal. We have added the section below to the discussion (Lines 355 – 364)

MSP1 is cleaved prior to merozoite release and thought to be presented on the merozoite surface in complex with other merozoite antigens (Kadekoppala and Holder, 2010). Nevertheless, we were still able to detect functional antibodies against the full-length protein. Other investigators have detected functional antibodies against subunits of MSP1 (Kana et al., 2019; Reiling et al., 2019). Taken together, this suggests that the epitopes responsible for inducing the types of functional antibodies measured against MSP1_{FL} are sufficiently exposed in vivo. Recently published work on vaccine-induced antibodies against MSP1_{FL} in malaria naïve individuals showed that functional antibodies

were also induced against the subunits, particularly p83 and p42 (Rosenkranz et al., 2023). We cautiously speculate that the impact of the processing on functional immunity as measured in these studies is minimal.

It is not entirely clear how participants were classified as treated febrile/non-febrile (was this done by blood smear microscopic examination?). Similarly, how PCR+ve/-ve groups were separated was not explained. A clear explanation of how participants were classified as one of these four groups from the original population would be really useful. This should go in at the start of the results since this terminology is used immediately with no background as to how the classification works.

This has been addressed and was recommended by all reviewers.

Similarly, in the Figure legends, the term parasite growth patterns is used in the Figure legends, but nowhere else. It would be very beneficial to the manuscript to explain why the groups were partitioned as they were and use similar nomenclature throughout.

This has been corrected in all the Figure legends.

The authors have concluded from the data in this paper that full length MSP1 is a superior antigen for measuring correlates of protection. However, a full comparison with functional antibody responses to subunits using these samples was not undertaken, which weakens this conclusion. I suggest that the limitations of the comparison made in this regard (mentioned in 2nd last paragraph: 'We were unable to assess the functional activity of antibodies directed against individual subunits and specific peptides within MSP1') and outstanding reasons that still support this claim, should be discussed and compared directly in more detail since this is a key premise for the conclusion that full length MSP1 is a superior correlate of immunity of subunits.

We concur... the main finding is really that full-length MSP1 induces function across a range of immune effectors in different Fc-IgG dependent assays. We have removed all reference to this being "superior" to the subunits for vaccine design because our data do not directly support this conclusion. This is reflected throughout the manuscript and in the restructuring of the discussion.

Have the authors considered whether the way full length protein is presented in these assays differs from that of the subunits and whether this could also have a positive impact on the immune measures?

We did not consider this as we did not have sufficient antigen to test the subunits in functional assays. Moreover, our full-length antigen was expressed in mammalian cells while the subunits were expressed in E.coli. This likely introduced a level of noise, noted in the higher background reactivity we detected in ELISA assays with the subunit proteins. In a separate study, functional antibodies against E.coli expressed full-length and subunit antigens found no evidence that either had an advantage over the other. Our conclusions are based on laboratory assays which may not accurately reflect the true situation in vivo. We've added a few statements to this effect in the discussion (Lines 365 – 367).

Minor comments

- First section of results "anti-MSP1FL antibody levels were higher in non-treated versus treated CHMI-SIKA volunteers" the last paragraph refers to Figure S2, this needs to be changed to Figure S3.

The supplementary figures have been correctly labelled as S1 and S2

- Introduction:

'In other studies, antibodies targeting MSP1 have been shown to promote several mechanisms...'

Suggest change to

'In other studies, antibodies targeting regions of MSP1 have been shown to promote several mechanisms....'.

The phrasing for this has changed – Please see the end of line 68

-Results:

'We categorized the level of Fc-mediated functions of the study participants as either high or low (coded 1 or 0, respectively) based on function specific thresholds. We then summed the breadth scores across the five functions.'

Suggest changing to.

'We categorized the level of the 5 Fc-mediated functions for the study participants as either high or low (coded 1 or 0, respectively) based on function specific thresholds. We then summed the breadth scores across the five functions.'

This has been corrected, addressed above.

-By convention, all measurement values should have a space before the SI units.

This has been corrected throughout the methods. We retained % in the format n%. We hope this is agreeable.

March 20, 2024

RE: Life Science Alliance Manuscript #LSA-2023-01910R

Prof. Faith Hope Among'in Osier
Imperial College London
Department of Life Science
Sir Alexander Fleming Building
Exhibition Road, South Kensington
London SW7 2AZ
United Kingdom

Dear Dr. Osier,

Thank you for submitting your revised manuscript entitled "Full-length MSP1 is a major target of protective immunity after controlled human malaria infection". We would be happy to publish your paper in Life Science Alliance pending final revisions necessary to meet our formatting guidelines.

- please be sure that the authorship listing and order is correct and match in our system
- please upload all figure files as individual ones, including the supplementary figure files; all figure legends should only appear in the main manuscript file
- please add the Twitter handle of your host institute/organization as well as your own or/and one of the authors in our system
- please upload a clean version of the manuscript file without the track changes
- please use the [10 author names et al.] format in your references (i.e., limit the author names to the first 10)
- please rename legends for supplementary figures in the manuscript text to correspond with the actual figures
- please add callouts for Figure S1A-B to your main manuscript text

A. FINAL FILES:

B. MANUSCRIPT ORGANIZATION AND FORMATTING:

Sincerely,

Reviewer #2 (Comments to the Authors (Required)):

This is a revised form of this manuscript in which the authors have addressed adequately all the concerns expressed by this reviewer. This manuscript makes a strong case for the significance of strong immune responses to MSP1 being of probable significance to protection from acute malaria. Overinterpretation and overstatement, which was an issue in the original draft, has been brought under control, strengthening the manuscript overall.

Reviewer #3 (Comments to the Authors (Required)):

The authors have addressed my concerns and I have no further comments.

May 8, 2024

RE: Life Science Alliance Manuscript #LSA-2023-01910RR

Prof. Faith Hope Among'in Osier
Imperial College London
Department of Life Science
Sir Alexander Fleming Building
Exhibition Road, South Kensington
London SW7 2AZ
United Kingdom

Dear Dr. Osier,

Thank you for submitting your Research Article entitled "Full-length MSP1 is a major target of protective immunity after controlled human malaria infection". It is a pleasure to let you know that your manuscript is now accepted for publication in Life Science Alliance. Congratulations on this interesting work.

DISTRIBUTION OF MATERIALS:

Again, congratulations on a very nice paper. I hope you found the review process to be constructive and are pleased with how the manuscript was handled editorially. We look forward to future exciting submissions from your lab.

Sincerely,
